# Transformers are Universal In-context Learners

**Takashi Furuya**[1]     **Maarten V. de Hoop**[2]     **Gabriel Peyré**[3]

[1]Doshisha Univ. `takashi.furuya0101@gmail.com`
[2]Rice Univ. `mvd2@rice.edu`
[3]CNRS, ENS, PSL Univ. `gabriel.peyre@ens.fr`

## Abstract

Transformers are deep architectures that define "in-context mappings" which enable predicting new tokens based on a given set of tokens (such as a prompt in NLP applications or a set of patches for a vision transformer). In this work, we study in particular the ability of these architectures to handle an arbitrarily large number of context tokens. To mathematically, uniformly address their expressivity, we consider the case that the mappings are conditioned on a context represented by a probability distribution of tokens which becomes discrete for a finite number of these. The relevant notion of smoothness then corresponds to continuity in terms of the Wasserstein distance between these contexts. We demonstrate that deep transformers are universal and can approximate continuous in-context mappings to arbitrary precision, uniformly over compact token domains. This result implies, as a special case, that transformers are universal approximators for continuous permutation-invariant mappings over a fixed number of tokens. It also establishes the universal approximation capability of transformers for certain in-context learning tasks, demonstrating in particular their ability to perform regression within context. A key aspect of our results, compared to existing findings, is that for a fixed precision, a single transformer can operate on an arbitrary (even infinite) number of tokens. Additionally, it operates with a fixed embedding dimension of tokens (this dimension does not increase with precision) and a fixed number of heads (proportional to the dimension). The use of MLPs between multi-head attention layers is also explicitly controlled. We consider both unmasked attentions (as used for the vision transformer) and masked causal attentions (as used for NLP and time series applications). We tackle the causal setting leveraging a space-time lifting to analyze causal attention as a mapping over probability distributions of tokens.

## 1 Introduction

Transformers have revolutionized the field of machine learning with their powerful attention mechanisms as introduced by Vaswani et al. (2017). The exceptional performance and expressivity of large-scale transformers have been empirically well established for both NLP (Brown et al., 2020) and vision applications (Dosovitskiy et al., 2020). One key property of these architectures is their ability to leverage contexts of arbitrary length, which enables the parameterization of "in context" mappings with an arbitrarily large complexity. In this paper, we present a rigorous formalism to model inputs and the associated context with an arbitrarily large number of tokens, defining a notion of continuity that enables the analysis of their expressivity.

**Universality, from neural networks to neural operators.** Multilayer Perceptrons (MLP) with two layers are universal approximators, as shown decades ago in Cybenko (1989); Hornik et al. (1989), with a comprehensive review in Pinkus (1999). The significance of depth in enhancing expressivity is explored in Hanin & Sellke (2017); Yarotsky (2017). These results have been extended to cover a variety of architectural constraints on the networks, for instance, invoking weight sharing in Convolutional Neural Networks (CNN) (Zhou, 2020) and skip connections in ResNets (Cuchiero

et al., 2020; Tabuada & Gharesifard, 2022). It is also possible to design equivariant architectures, in particular for graph neural networks (Kratsios & Papon, 2022; Keriven & Peyré, 2019; Xu et al., 2018) and neural networks operating on sets of points (Qi et al., 2017; De Bie et al., 2019). The connection between transformers and graph neural networks is exposed in Müller et al. (2023). Here, we take a different point of view, with transformers operating on probability distributions rather than on sets of points. Related to this setup are extensions of neural networks acting in finite-dimensional vector spaces to infinite-dimensional function spaces resulting in the notion of neural operators (Kovachki et al., 2023), the universality of which is studied in Furuya et al. (2023). Neural operators can be generalized to cope with data in metric spaces, addressing topological obstructions, in Kratsios et al. (2023b).

**Mathematical modeling of transformers.** It is now customary to describe transformers as performing "in context" prediction, which means that it maps token to token, while this map depends on a set of previously seen tokens. The size of this context might be very long, possibly arbitrarily long, which is the focus of this article. The ability of trained transformers to effectively perform in-context computation has been supported by both empirical studies (von Oswald et al., 2023) and theoretical results (Ahn et al., 2024; Mahankali et al., 2023; Sander et al., 2024; Zhang et al., 2023) on simplified architectures (typically linear attention) and specific data generation processes.

To make a rigorous analysis of arbitrarily long token lengths, and also to describe a "mean field" limit of an infinite number of tokens, it is convenient to view attention as operating over probability distributions of tokens (Vuckovic et al., 2020; Sander et al., 2022). The smoothness (Lipschitz continuity) of the resulting attention layers is analyzed in Castin et al. (2024). Deep transformers (with the residual connection) can be described by a coupled system of particles evolving across the layers. The analysis of the clustering properties of such an evolution is studied in Geshkovski et al. (2023a;b).

**Universality of transformers.** Yun et al. (2019) provides, to the best of our knowledge, the most detailed account of the universality of transformers. The authors rely on shallow transformers with only 2 heads but require that the transformers operate over an embedding dimension which grows with the number of tokens. This result is refined in Nath et al. (2024) which highlights the difficulty of attention mechanisms to capture smooth functions. Our focus is different, since we consider deep transformers with a fixed embedding dimension, but which are universal for an arbitrary number of tokens.

We note that there exist variations over the original transformer's architecture which enjoys universality results, for instance, the Sumformer (Alberti et al., 2023) and stochastic deep networks (De Bie et al., 2019), which also requires an embedding dimension that grows with the number of tokens. We furthermore mention the introduction of probabilistic transformers (Kratsios et al., 2023a) which can approximate embeddings of metric spaces. The work of Agrachev & Letrouit (2024) provides an abstract universal interpolation result for equivariant architectures under genericity conditions, but it is not known whether there exist generic attention maps.

While this is not directly related to our results, a line of works studies the expressivity of transformers when operating on a discrete set of tokens as formal systems (Chiang et al., 2023; Merrill & Sabharwal, 2023; Strobl et al., 2024; Elhage et al., 2021). Another line of work studies the impact of positional encoding on their expressivity (Luo et al., 2022).

## 1.1 OUR CONTRIBUTIONS

Our work provides a rigorous formalization of transformer expressivity and continuity as operating over the space of probability distributions. The main mathematical results is the universality presented in Theorems 1 and 2, respectively, for the unmasked and the masked settings. Our approach effectively handles an arbitrary number of tokens and leverages deep architectures without requiring arbitrary width. The embedding dimension and the number of heads are proportional to the dimension of the input tokens and are independent of precision. It is interesting to note that the masked setting requires some stronger regularity hypothesis on the contexts, namely that they are Wasserstein-Lipschitz with respect to time, which is needed to cope with the constraint of causality in the relevant mappings.

## 1.2 NOTATION

For a natural number $N \in \mathbb{N}$, we denote by $[N] := \{1, ..., N\}$. For vector $x \in \mathbb{R}^d$, the Euclidean norm of $x$ is denoted by $|x|$. For two vectors $x, y \in \mathbb{R}^d$, the Euclidean inner product of $x$ and $y$ is denoted by $\langle x, y \rangle$ and the component-wise multiplication of $x$ and $y$ is denoted by $x \odot y$. The vector $\mathbf{1}_d$ is the vector of dimension $d$ with all coordinates equal to 1, that is, $\mathbf{1}_d := (1, ..., 1) \in \mathbb{R}^d$. We denote by $\mathcal{P}(\Omega)$ the space of probability measures on $\Omega$, and denote by $\mathcal{C}(\Omega)$ the space of continuous functions from $\Omega$ to $\mathbb{R}$, where $\Omega \subset \mathbb{R}^d$ is a compact domain for tokens' embeddings. In what follows, we frequently utilize notions such as the push-forward operator $T_\sharp$, weak* topology (denoted by the convergence $\rightharpoonup^*$), and Wasserstein distance $W_p$ for $1 \leq p < +\infty$. For further details on these notions, we refer to Appendix A.

## 2 MEASURE-THEORETIC IN-CONTEXT MAPPINGS

Transformers are defined by alternating multi-head attention layers (which compute interactions between tokens), MLP and normalization layers (which operate independently over each token). For the sake of simplicity, we omit normalization in the following analysis. We first recall their definition and then explain how they can be equivalently re-written using in-context mappings. This definition provides new insights and can also be generalized to an infinite number of tokens encoded in a probability measure.

### 2.1 ATTENTION AS IN CONTEXT MAPPINGS ON TOKEN ENSEMBLES

**Classical definition.** A set of $n$ tokens, $x_i \in \mathbb{R}^{d_{\text{tok}}}$, is denoted by $X = (x_i)_{i=1}^n \in \mathbb{R}^{d_{\text{tok}} \times n}$. An attention head maps these $n$ tokens to the same number $n$ of tokens in $\mathbb{R}^{d_{\text{head}}}$ through

$$\forall X \in \mathbb{R}^{d_{\text{tok}} \times n}, \quad \text{Att}_\theta(X) := VX \, \text{SoftMax}(X^\top Q^\top K X / \sqrt{k}) \in \mathbb{R}^{d_{\text{head}} \times n},$$

where the parameters are the (Key, Query, Value) matrices $\theta := (K, Q, V) \in \mathbb{R}^{k \times d_{\text{tok}}} \times \mathbb{R}^{k \times d_{\text{tok}}} \times \mathbb{R}^{d_{\text{head}} \times d_{\text{tok}}}$. Here, the (possibly masked) SoftMax function operates in a row-wise manner:

$$\forall Z \in \mathbb{R}^{n \times n}, \quad \text{SoftMax}(Z) := \left( \frac{M_{i,j} e^{Z_{i,j}}}{\sum_{\ell=1}^n M_{i,\ell} e^{Z_{i,\ell}}} \right)_{i,j=1}^n \in \mathbb{R}_+^{n \times n},$$

where $M_{i,j} = 1$ for the unmasked setting (bidirectional encoding transformers) and $M_{i,j} = 1_{j \leq i}$ in the masked setting (causal decoding transformers). Multiple heads with different parameters $\theta := (W^h, \theta^h)_{h=1}^H$ are combined in a linear way in a multi-head attention

$$\text{MAtt}_\theta(X) := X + \sum_{h=1}^H W^h \, \text{Att}_{\theta^h}(X), \tag{1}$$

where $W^h \in \mathbb{R}^{d_{\text{tok}} \times d_{\text{head}}}$ and $\theta^h := (K^h, Q^h, V^h)$. In the following, we denote the various dimensions of a multi-head attention layer by: $d_{\text{tok}}(\theta), d_{\text{head}}(\theta)$ for the token and head dimensions, respectively, and $k(\theta)$ for the key/query dimensions, and $H(\theta)$ for the number of heads.

**In-context mappings form.** For the unmasked setting, the mapping $X \mapsto \text{MAtt}_\theta(X)$ can be re-written as the application of an "in context" function $G_\theta(X, \cdot)$ to each token,

$$x_i \mapsto G_\theta(X, x_i) \quad \text{i.e.} \quad \text{MAtt}_\theta(X) = (G_\theta(X, x_i))_{i=1}^n,$$

where the in-context mapping is, $\forall (X, x) \in \mathbb{R}^{d_{\text{tok}} \times n} \times \mathbb{R}^{d_{\text{tok}}}$,

$$G_\theta(X, x) := x + \sum_{h=1}^H W^h \sum_{j=1}^n \frac{\exp\left( \frac{1}{\sqrt{k}} \langle Q^h x, K^h x_j \rangle \right)}{\sum_{\ell=1}^n \exp\left( \frac{1}{\sqrt{k}} \langle Q^h x, K^h x_\ell \rangle \right)} V^h x_j. \tag{2}$$

In the masked setting, due to the lack of permutation equivariance, it is required to track also the index $i$ of the token. The mapping $X \mapsto \text{MAtt}_\theta(X)$ can then be re-written as $\text{MAtt}_\theta(X) = (G_\theta(X, x_i, i))_{i=1}^n$ where the in-context mapping is,

$$G_\theta(X, x, i) := x + \sum_{h=1}^H W^h \sum_{j=1}^i \frac{\exp\left( \frac{1}{\sqrt{k}} \langle Q^h x, K^h x_j \rangle \right)}{\sum_{\ell=1}^i \exp\left( \frac{1}{\sqrt{k}} \langle Q^h x, K^h x_\ell \rangle \right)} V^h x_j. \tag{3}$$

Here, the terminology "in context" refers to the fact that $G_\theta(X, \cdot)$ depends on the tokens $X$ themselves, and can thus be seen as a parametric map that is modified for each token depending on its interactions with the other tokens. While this re-writing is equivalent to the original one, it highlights the fact that transformers define spatial mappings. This also allows us to clearly state the associated mathematical question at the core of this paper, which is the approximation of arbitrary in-context mappings by (compositions of) such parametric maps. Another interest in this reformulation is that it enables the definition of generalized attention operating over a possibly infinite number of tokens, as explained in Section 2.2.

**Composition of in-context mappings.** A transformer (ignoring normalization layers at this moment) is a composition of $L$ attention layers and Multi-Layer Perceptrons (MLP):

$$\text{MLP}_{\xi_L} \circ \text{MAtt}_{\theta_L} \circ \ldots \circ \text{MLP}_{\xi_1} \circ \text{MAtt}_{\theta_1}. \tag{4}$$

Here, the $\text{MLP}_\xi$ functions process each token independently from one another:

$$\text{MLP}_\xi(X) = (F_\xi(x_i))_{i=1}^n,$$

i.e., they are "context-free" mappings (in the above notation, $F_\xi(X, x) = F_\xi(x)$), while the attention maps, $G_\theta(X, \cdot)$ depend on the context $X$.

On the level of in-context mappings, the composition of layers in (4) induces a new "in-context" composition rule, which we denote by $\diamond$:

$$(G_2 \diamond G_1)(X, x) := G_2(X_1, G_1(X, x)) \text{ where } X_1 := (G_1(X, x_i))_{i=1}^n, \tag{5}$$

for the unmasked case, and

$$(G_2 \diamond G_1)(X, x, i) := G_2(X_1, G_1(X, x, i), i) \text{ where } X_1 := (G_1(X, x_i, i))_{i=1}^n, \tag{6}$$

for the masked case. This rule can be applied whether $G_1(X, \cdot)$ or $G_2(X, \cdot)$ depends on the context $X$ or not (such as for the $F_\xi$ mappings above). Using this rule, the transformer's definition in (4) translates into a composition of in-context and context-free maps, i.e.,

$$F_{\xi_L} \diamond G_{\theta_L} \diamond \ldots \diamond F_{\xi_1} \diamond G_{\theta_1}. \tag{7}$$

The core question this paper addresses is the uniform approximation of a continuous (in a suitable topology) in-context maps $(X, x) \mapsto G(X, x)$ or $(X, x, i) \mapsto G(X, x, i)$ by transformers' in-context mappings of the form of (4), with clear control of the dimensions and the number of heads involved in the different layers. The main originality of our approach is that we aim to do so for an arbitrary number $n$ of tokens, as we now explain.

## 2.2 Measure-theoretic in-context mappings: Unmasked setting

A first key observation is that the definition in (2) makes sense irrespective of the number $n$ of tokens. The second key observation is that, in the un-masked case, $M_{i,j} = 1$, the attention mapping is permutation equivariant. To make this more explicit, and also handle the limit of an infinite number of tokens, we represent a set $X$ of tokens using a probability distribution $\mu \in \mathcal{P}(\mathbb{R}^{d_{\text{tok}}})$ over $\mathbb{R}^{d_{\text{tok}}}$. A finite number of tokens is encoded using a discrete empirical measure,

$$\mu = \frac{1}{n} \sum_{i=1}^n \delta_{x_i} \in \mathcal{P}(\mathbb{R}^{d_{\text{tok}}}). \tag{8}$$

This encoding is not only for notional convenience, it also allows us to define clearly a correct notion of smoothness for the in-context mappings. This smoothness corresponds to the displacement of the tokens and is quantified through the optimal transport distance as presented in Section 1.2. This enables us to compare context with different sizes and, for instance, to compare a set of tokens with a large (but finite) $n$ to a continuous distribution. The in-context mapping in (2) is now defined as, $\forall(\mu, x) \in \mathcal{P}(\mathbb{R}^{d_{\text{tok}}}) \times \mathbb{R}^{d_{\text{tok}}}$,

$$\Gamma_\theta(\mu, x) := x + \sum_{h=1}^H W^h \int \frac{\exp\left(\frac{1}{\sqrt{k}}\langle Q^h x, \, K^h y\rangle\right)}{\int \exp\left(\frac{1}{\sqrt{k}}\langle Q^h x, \, K^h z\rangle\right) \mathrm{d}\mu(z)} V^h y \, \mathrm{d}\mu(y). \tag{9}$$

The discrete case is contained in this more general definition in the sense that

$$\forall X = (x_i)_{i=1}^n, \quad G_\theta(X, x) = \Gamma_\theta\Big(\frac{1}{n}\sum_{i=1}^n \delta_{x_i}, x\Big).$$

In the following, we will invoke, whenever convenient, the following slight abuse of notation,

$$\Gamma_\theta(\mu, x) = \Gamma_\theta(\mu)(x),$$

so that $\Gamma_\theta(\mu) : \mathbb{R}^{d_{\text{tok}}} \to \mathbb{R}^{d_{\text{tok}}}$ defines a map between Euclidean spaces. Using this general definition, the attention map $X \mapsto \text{MAtt}_\theta(X)$ can be rewritten as displacing the tokens' positions, which corresponds to applying a push-forward to the measure as defined in (22),

$$\mu \in \mathcal{P}(\mathbb{R}^{d_{\text{tok}}}) \longmapsto \Gamma_\theta(\mu)_\sharp\mu \in \mathcal{P}(\mathbb{R}^{d_{\text{tok}}}).$$

This formulation of transformers as a mapping between probability measures was introduced in Sander et al. (2022) and also used in Castin et al. (2024) to prove a convergence result of deep transformers. We re-use it here but put emphasis on the in-context mapping itself, which is the object of interest of this paper (rather than on studying the mapping between measures).

**Composition of in-context unmasked measure-theoretic mappings.** The definition of composition in (5) generalizes to the measure-theoretic setting in the unmasked setting as

$$(\Gamma_2 \diamond \Gamma_1)(\mu, x) := \Gamma_2(\mu_1, \Gamma_1(\mu, x)), \quad \text{where} \quad \mu_1 := \Gamma_1(\mu)_\sharp\mu, \tag{10}$$

i.e., $(\Gamma_2 \diamond \Gamma_1)(\mu) = \Gamma_2(\mu_1) \circ \Gamma_1(\mu)$. Transformers operating over an arbitrary (possibly infinite) number of tokens are then obtained by replacing the original definition of (7) by

$$F_{\xi_L} \diamond \Gamma_{\theta_L} \diamond \ldots \diamond F_{\xi_1} \diamond \Gamma_{\theta_1}. \tag{11}$$

Here, the $F_\xi$ are "context-free" MLP mappings, i.e., $F_\xi(\mu, x) = F_\xi(x)$ is independent of $\mu$. It is important to keep in mind that when restricted to finite discrete empirical measures of the form of (8), definitions in (7) and in (11) coincide. Our theory encompasses classical transformers as well as their "mean field" limits operating over arbitrary measures.

## 2.3 MEASURE-THEORETIC IN-CONTEXT MAPPINGS: MASKED SETTING

In the masked setting (for NLP or time series applications), $M_{i,j} = 1_{j \leq i}$, the attention mappings are not any more permutation equivariant. To restore this invariance, and be able to write in-context mappings using measures, we introduce a space-time lifting so that the input tokens are of the form $\{x_i, t_i\}_{i=1}^n$, where $t_i \in [0, 1]$. For instance, assuming an upper bound, $N$, on the number of tokens, one can use $t_i = i/N$, but it is also possible to assume that the $t_i$ are positioned arbitrarily in $[0, 1]$, which enables considering an arbitrarily large (and even infinite) number of tokens.

We thus let the context be encoded as a space-time measure $\mu \in \mathcal{P}(\mathbb{R}^{d_{\text{tok}}} \times [0, 1])$. Similarly to Castin et al. (2024, Definition 2.6), we introduced the in-context map, $\forall (x, t) \in \mathbb{R}^{d_{\text{tok}}} \times [0, 1]$,

$$\Gamma_\theta(\mu, x, t) := x + \sum_{h=1}^H W^h \int \frac{\exp\left(\frac{1}{\sqrt{k}}\langle Q^h x, K^h y\rangle\right) 1_{[0,t]}(r)}{\int \exp\left(\frac{1}{\sqrt{k}}\langle Q^h x, K^h z\rangle\right) 1_{[0,t]}(s) \, d\mu(z, s)} V^h y \, d\mu(y, r), \tag{12}$$

where $1_{[0,t]}(s)$ is a masking function that is 1 if $0 \leq s \leq t$ and 0 otherwise. For a finite number of tokens, using a discrete measure, $\mu = \frac{1}{n}\sum_{i=1}^n \delta_{(x_i, i/n)}$, one retrieves the initial definition in (3) in the masked case. The space-time lifting can incorporate positional information, however, recent positional encoding methods, such as Rotary Positional Embedding (RoPE) (Kazemnejad et al., 2024), are not included in the current formulation. This extension is left for future work.

**Composition of in-context masked measure-theoretic mappings.** The composition rule in the masked setting is similar to the one in the unmasked setting (cf. (10)), except that the time position of the token is kept unchanged while the push forward acts in space,

$$(\Gamma_2 \diamond \Gamma_1)(\mu, x, t) := \Gamma_2(\mu_1, \Gamma_1(\mu, x, t), t), \quad \text{where} \quad \mu_1 := (\Gamma_1(\mu), \text{Id}_\mathbb{R})_\sharp\mu. \tag{13}$$

Here, $(\Gamma_1(\mu), \text{Id}_\mathbb{R}) : (x, t) \in \mathbb{R}^{d_{\text{tok}}+1} \to (\Gamma_1(\mu)(x, t), t) \in \mathbb{R}^{d_{\text{tok}}+1}$ (the time is kept unchanged). Equipped with this definition, one retrieves the composition rule in (6) when the measure $\mu$ is discrete.

## 3 Universality in the unmasked case

### 3.1 Statement of the result and discussion

Our first main result is the following universal approximation theorem for unmasked in-context mappings.

**Theorem 1.** *Let $\Omega \subset \mathbb{R}^d$ be a compact set and $\Lambda^\star : \mathcal{P}(\Omega) \times \Omega \to \mathbb{R}^{d'}$ be continuous, where $\mathcal{P}(\Omega)$ is endowed with the weak$^*$ topology. Then for all $\varepsilon > 0$, there exist $L$ and parameters $(\theta_\ell, \xi_\ell)_{\ell=1}^L$, such that*

$$\forall (\mu, x) \in \mathcal{P}(\Omega) \times \Omega, \quad |F_{\xi_L} \diamond \Gamma_{\theta_L} \diamond \ldots \diamond F_{\xi_1} \diamond \Gamma_{\theta_1}(\mu, x) - \Lambda^\star(\mu, x)| \leq \varepsilon,$$

*with $d_{\mathrm{tok}}(\theta_\ell) \leq d + 3d'$, $d_{\mathrm{head}}(\theta_\ell) = k(\theta_\ell) = 1$, $H(\theta_\ell) \leq d'$.*

Appendix D demonstrates that this result can be applied to two key settings: permutation-invariant mappings over a fixed number of tokens and in-context learning of regression operators.

The two strengths of the result above are (i) the approximating architecture performs the approximation independently of $n$ (it even works for an infinite number of tokens), and (ii) the number of heads and the embedding dimension do not depend on $\varepsilon$. Moreover, its proof technique is notable, particularly because, unlike classical MLPs with cosine activation functions, shallow attention architectures lack an algebraic structure as these cannot be multiplied. To address this, we rely on depth to accommodate this limitation.

A weakness is that the theorem is "non-quantitative", meaning that that we have no explicit control over the dependency of the number of MLP parameters $\xi_\ell$ on $\varepsilon$. A limitation of our proof technique is that the number of heads grows proportionally to the output dimension while each head only outputs a scalar $d_{\mathrm{head}}(\theta_\ell) = 1$. Obtaining a better balance between these two parameters is an interesting problem. As explained in the proof, these MLPs approximate a real-valued squaring operator $\mathbb{R} \ni a \mapsto a^2$, so we expect this dependency to be well-behaved in common situations; however, our construction does not provide any a priori bound on how the magnitude of the tokens grows through the layers. The main hypothesis of Theorem 1 is that the underlying map, $\Lambda^\star$, is a continuous map for the weak$^*$ topology over measures (see Section 1.2 for some background). However, this setting might not be a proper one for conducting further quantitative studies; we leave these for future work.

### 3.2 Proof of Theorem 1

We first consider "elementary" in-context mappings, which map $(x, \mu)$ to a real variable

$$\gamma_\lambda(\mu, x) := \langle x, a \rangle + b + \int \frac{e^{c(\langle x, a \rangle + b)(\langle y, a \rangle + b)} v\left(\langle a, y \rangle + b\right)}{\int e^{c(\langle x, a \rangle + b)(\langle z, a \rangle + b)} \mathrm{d}\mu(z)} \, \mathrm{d}\mu(y), \tag{14}$$

where $\lambda := (a, b, c, v) \in \mathbb{R}^d \times \mathbb{R} \times \mathbb{R} \times \mathbb{R}$. These elementary mappings are built by composing an affine scalar-valued MLP with a single-head attention (with skip connection) as in (9), operating in 1-D. Indeed, defining $F_\xi(x) = \langle a, x \rangle + b \in \mathbb{R}$ as an affine MLP, where $\xi = (a, b)$, we have

$$\gamma_\lambda(\mu, x) = (\Gamma_\theta \diamond F_\xi)(\mu, x),$$

where $\theta = (k, q, v) \in \mathbb{R}^3$ (recalling that this attention operates in 1-D), and we let $c = qk$.

We now define $\mathcal{A}$, the algebra spanned by these elementary functions:

$$\mathcal{A} := \left\{ \mathcal{P}(\Omega) \times \Omega \ni (\mu, x) \mapsto \sum_{n=1}^N \gamma_{\lambda_{1,n}}(\mu, x) \odot \cdots \odot \gamma_{\lambda_{T,n}}(\mu, x) \in \mathbb{R} : N, T \in \mathbb{N} \right\}.$$

The first main ingredient of the proof is to show that this algebra is dense. Elements of this algebra are sums of products of elementary functions, which are often referred to as "cylindrical functions".

**Proposition 1.** *$\mathcal{A}$ is dense in the space of (weak$^*$ $\times \ell^2$)-continuous functions from $\mathcal{P}(\Omega) \times \Omega$ to $\mathbb{R}$.*

*Proof.* We apply the Stone-Weierstrass theorem. First, we note that $\mathcal{P}(\Omega) \times \Omega$ is compact for the (weak$^*$ $\times \ell^2$) topology (Aliprantis & Border, 2006, Theorem 15.11). We then check the three key hypotheses needed to apply the Stone-Weierstrass theorem:

1. The functions $\gamma_\lambda$ are continuous because the denominator $\int e^{c(\langle x, a\rangle + b)(\langle z, a\rangle + b)}\mathrm{d}\mu(z)$ in the elementary mapping is not always zero for any $\mu \in \mathcal{P}(\Omega)$ and $x \in \Omega$.

2. When setting $a = 0, b = v = 1$, one has that $\gamma_\lambda(\mu, x) = 1$ is the constant function.

3. We need to show that the set $(\gamma_\lambda)_\lambda$ separates points, which is more challenging.

For the last one, we need to show that if

$$\forall \lambda, \gamma_\lambda(\mu, x) = \gamma_\lambda(\mu', x'), \tag{15}$$

then $(\mu, x) = (\mu', x')$. First setting $v = 0$, this implies that $\langle x, a\rangle = \langle x', a\rangle$ for all $a \in \mathbb{R}^d$ and, hence, $x = x'$. Then, setting $b = 1 - \langle a, x\rangle$, (15) reads $L(\mu)(a, c) = L(\mu')(a, c)$ where we defined a generalized Laplace-like transform

$$L(\mu)(a, c) := \int \frac{e^{c\langle a, y\rangle}\langle a, y\rangle}{\int e^{c\langle a, z\rangle}\mathrm{d}\mu(z)}\,\mathrm{d}\mu(y). \tag{16}$$

We conclude that $\mu = \mu'$ using the following key lemma. $\qquad\square$

**Lemma 1.** *The map $\mu \mapsto L(\mu)$ defined in (16) is injective.*

See Appendix B.1 for a detailed proof. To approximate vector-valued in-context mappings, we use the previous algebra of cylindrical functions along each dimension. Since the elementary mapping, $\gamma_\lambda$, is built by the composing an affine MLP and single-head attention, we arrive at the following lemma.

**Lemma 2.** *For any $\varepsilon > 0$, there exist $T, N \in \mathbb{N}$ and $(\tilde{\theta}_{t,n}, \tilde{\xi}_{t,n})_{t\in[T],n\in[N]}$ such that*

$$\forall(\mu, x) \in \mathcal{P}(\Omega) \times \Omega, \quad |G(\mu, x) - \Lambda^\star(\mu, x)| \leq \varepsilon,$$

$$\textit{where} \quad G(\mu, x) := \sum_{n=1}^{N}(\Gamma_{\tilde{\theta}_{1,n}} \diamond F_{\tilde{\xi}_{1,n}})(\mu, x) \odot \cdots \odot (\Gamma_{\tilde{\theta}_{T,n}} \diamond F_{\tilde{\xi}_{T,n}})(\mu, x), \tag{17}$$

*with $d_{\mathrm{tok}}(\tilde{\theta}_{t,n}) = d'$, $d_{\mathrm{head}}(\tilde{\theta}_{t,n}) = k(\tilde{\theta}_{t,n}) = 1$, $H(\tilde{\theta}_{t,n}) = d'$.*

See Appendix B.2 for the details of the proof. We finally need to approximate $G$ in (17) by a deep transformer of the form of (7). To do that, we furthermore approximate the component-wise multiplication maps, $(x, y) \in \mathbb{R}^{2d'} \mapsto x \odot y \in \mathbb{R}^{d'}$, in (17) by some MLPs. This way, we obtain the following lemma.

**Lemma 3.** *For any $\varepsilon > 0$, there exist $L$ and parameters $(\theta_\ell, \xi_\ell)_{\ell=1}^{L}$, such that*

$$\forall(\mu, x) \in \mathcal{P}(\Omega) \times \Omega, \quad |G(\mu, x) - F_{\xi_L} \diamond \Gamma_{\theta_L} \diamond \ldots \diamond F_{\xi_1} \diamond \Gamma_{\theta_1}(\mu, x)| \leq \varepsilon,$$

*with $d_{\mathrm{tok}}(\theta_\ell) \leq d + 3d'$, $d_{\mathrm{head}}(\theta_\ell) = k(\theta_\ell) = 1$, $H(\theta_\ell) \leq d'$.*

See Appendix B.3 for a detailed proof.

## 4 UNIVERSALITY IN THE MASKED CASE

### 4.1 CAUSAL MAPS, LIPSCHITZ CONTEXTS, AND MAIN RESULT

As before, $\Omega \subset \mathbb{R}^d$ is a compact set, and we define $\tilde{\Omega} := \Omega \times [0, 1]$ as the space-time domain. Throughout this section, $\bar{\mu} \in \mathcal{P}([0, 1])$ is the marginal with respect to only the time variable of some space-time measure, $\mu \in \mathcal{P}(\tilde{\Omega})$, i.e.,

$$\bar{\mu} := P_\sharp\mu \quad \text{where} \quad P : \tilde{\Omega} \ni (x, t) \mapsto t \in [0, 1]. \tag{18}$$

Approximation in the masked setting is more subtle than in the unmasked setting because causal attentions are typically less regular due to the masking. To cope with this difficulty, we impose additional smoothness constraints on the context, which still allow for an arbitrary number of tokens.

**Definition 1** (Lipschitz contexts). *A map $t \in [0,1] \mapsto \mu(\cdot|t) \in \mathcal{P}(\Omega)$ is $C$-Lipschitz if*

$$\forall (s,t) \in [0,1]^2, \quad W_2(\mu(\cdot|s), \mu(\cdot|t)) \leq C|s-t|.$$

*The set of space-time $C$-Lipschitz measures is*

$$\mathrm{Lip}_C(\tilde{\Omega}) := \{\mu \in \mathcal{P}(\tilde{\Omega}) \,:\, \exists \mu(\cdot|t) \text{ s.t. } \mu(x,s) = \mu(x|s)\bar{\mu}(s) \text{ and } \mu(\cdot|t) \text{ is } C\text{-Lipschitz}\}.$$

*The conditional measure $t \mapsto \mu(\cdot|t) \in \mathcal{P}(\Omega)$ is any valid disintegration of $\mu$ against the marginal $\bar{\mu}$, and must be $C$-Lipschitz. We also define, $\forall \sigma \in (0,1)$,*

$$\mathrm{Lip}_C^\sigma(\tilde{\Omega}) := \{\mu \in \mathrm{Lip}_C(\tilde{\Omega}) \,:\, \bar{\mu}(\{0\}) \geq \sigma\}.$$

It is worth noting that these conditions are automatically satisfied by a discrete measure, $\mu = \frac{1}{n}\sum_{i=1}^n \delta_{(x_i,t_i)}$, with distinct times $\delta := \min_{i \neq j} |t_i - t_j| > 0$ (using $C = \mathrm{Radius}(\Omega)/\delta$). More generally, if $t \in [0,1] \to \phi(t) \in \Omega$ is $C$-Lipschitz and $\nu \in \mathcal{P}([0,1])$, then $\mu = \phi_\sharp \nu \in \mathrm{Lip}_C(\tilde{\Omega})$.

Masked attention can be conveniently re-expressed as operating over masked contexts which are defined as follows.

**Definition 2** (Masked measure). *For $\mu \in \mathrm{Lip}_C^\sigma(\tilde{\Omega})$, the masked probability measure $\mu_t \in \mathrm{Lip}_C^\sigma(\tilde{\Omega})$ is defined as*

$$\mu_t := \frac{1_{[0,t]}}{\bar{\mu}([0,t])} \cdot \mu. \tag{19}$$

Thanks to $\mu \in \mathrm{Lip}_C^\sigma(\tilde{\Omega})$, where the starting point of $\bar{\mu}$ is fixed as $0$ (i.e., $0 \in \mathrm{supp}(\bar{\mu})$), the masked probability measure $\mu_t$ is well-defined when $t \in (0,1]$. We note that we can define $\mu_t$ at $t = 0$ as the limit (in the weak$^*$ topology), and such a limit exists (See Lemma 10). Thus, the map $[0,1] \ni t \mapsto \mu_t \in \mathrm{Lip}_C^\sigma(\tilde{\Omega})$ is continuous (for the weak$^*$ topology).

In the masked setting, the aim is to approximate causal mappings, defined as follows, which are maps where the output of time $t$ only depends on tokens with smaller times.

**Definition 3** (Causal identifiable map). *A space-time in-context map $\Lambda$ is said to be causal if*

$$\forall (\mu, x, t) \in \mathrm{Lip}_C^\sigma(\tilde{\Omega}) \times \tilde{\Omega}, \quad \Lambda(\mu, x, t) = \Lambda(\mu_t, x, t). \tag{20}$$

*Such a map $\Lambda$ is said to be identifiable if for any context $\mu \in \mathrm{Lip}_C^\sigma(\tilde{\Omega})$,*

$$\mu_t = \mu_{t'} \quad \Rightarrow \quad \Lambda(\mu_t, \cdot, t) = \Lambda(\mu_{t'}, \cdot, t'). \tag{21}$$

By the construction, the masked attention map $\Gamma_\theta$, defined in (12), is a causal identifiable in-context map (See Lemma 11). Since the composition of such maps preserves both causality and identifiability (see Lemma 12), a deep transformer, formed by composing of causal identifiable in-context maps, remains these properties.

The following theorem mimics Theorem 1, but is restricted to approximating on Lipschitz contexts to cope with the causality constraint.

**Theorem 2.** *Let $\Lambda^\star$ be a continuous (where $\mathrm{Lip}_C^\sigma(\tilde{\Omega})$ is endowed with the weak$^*$ topology) and causal identifiable in-context mapping. Then, for all $\varepsilon > 0$, there exist $L$ and parameters $(\theta_\ell, \xi_\ell)_{\ell=1}^L$ such that*

$$\forall (\mu, x, t) \in \mathrm{Lip}_C^\sigma(\tilde{\Omega}) \times \tilde{\Omega}, \; |F_{\xi_L} \diamond \Gamma_{\theta_L} \diamond \ldots \diamond F_{\xi_1} \diamond \Gamma_{\theta_1}(\mu, x, t) - \Lambda^\star(\mu, x, t)| \leq \varepsilon,$$

*with $d_{\mathrm{tok}}(\theta_\ell) \leq d + 3d'$, $d_{\mathrm{head}}(\theta_\ell) = k(\theta_\ell) = 1$, $H(\theta_\ell) \leq d'$.*

**Remark 1** (Sharpness of the identifiability and Lipschitz hypotheses). *The masked approximation in Theorem 2 shares the same conclusion as the unmasked Theorem 1, but it requires stronger assumptions. In particular, the map $\Lambda^\star$ is assumed to be identifiable. This hypothesis is sharp and cannot be weakened: transformers define identifiable maps, and as proved in Lemma 13 identifiable maps can only approximate uniformly identifiable maps. Identifiability is also crucial since our proof technique involves recasting the approximation over $(\mu, x, t)$ as an approximation over a reduced space $(\mu_t, x)$. Another important assumption is that we restrict our approximation to Lipschitz contexts. This limitation is essential for ensuring that the set of masked contexts $\mu_t$ is compact, which allows us to apply the Stone-Weierstrass theorem.*

**Remark 2** (Fixing the time marginal). *We impose that contexts have Dirac masses at 0, namely $\bar{\mu}(\{0\}) \geq \sigma$. While this is not restrictive for discrete measures, this prevents for instance measures with density with respect to Lebesgues. It is possible to lift this constraint, and instead impose that the time marginal $\bar{\mu}$ is fixed, i.e. replace the set $\mathrm{Lip}_C^\sigma(\tilde{\Omega})$ by $\{\mu \in \mathrm{Lip}_C(\tilde{\Omega}) : \bar{\mu} = \nu\}$ for any $\nu \in \mathcal{P}([0,1])$ satisfying $0 \in \mathrm{supp}(\nu)$. One can check that the proof of Theorem 2 carries over to this setting with minor modifications.*

## 4.2 PROOF OF THEOREM 2

To translate into the setting that can exploit the Stone-Weierstrass theorem, we first introduce the following operation, which can be defined for any probability measure.

**Definition 4** (Reduced mapping). *For $\Lambda : \mathcal{P}(\tilde{\Omega}) \times \tilde{\Omega} \to \mathbb{R}^{d'}$, we define the reduced map $\bar{\Lambda}$, which takes two argument $(\mu, x)$, as*

$$\forall (\mu, x) \in \mathcal{P}(\tilde{\Omega}) \times \tilde{\Omega}, \quad \bar{\Lambda}(\mu, x) := \Lambda(\mu, x, e(\bar{\mu})),$$

*where $e(\bar{\mu})$ is the end point of $\mathrm{supp}(\bar{\mu})$, that is, $e(\bar{\mu}) := \max\{r \in \mathrm{supp}(\bar{\mu})\} \in [0,1]$.*

We introduce the reduced space on which we consider the approximation,

$$\mathcal{X}_C^\sigma := \{(\mu_t, x) : \mu \in \mathrm{Lip}_C^\sigma(\tilde{\Omega}), \ x \in \Omega, \ t \in [0,1]\}.$$

**Lemma 4.** *Let $\Lambda : \mathrm{Lip}_C^\sigma(\tilde{\Omega}) \times \tilde{\Omega} \to \mathbb{R}^{d'}$ be a causal identifiable in-context mapping defined in Definition 3. Then the following holds true.*

*(i) For any $(\mu, x, t) \in \mathrm{Lip}_C^\sigma(\tilde{\Omega}) \times \tilde{\Omega}, \quad \Lambda(\mu, x, t) = \bar{\Lambda}(\mu_t, x)$.*

*(ii) If $\Lambda$ is continuous, then the reduced map of $\mathcal{X}_C^\sigma \ni (\mu_t, x) \mapsto \bar{\Lambda}(\mu_t, x)$ is (weak$^*$ $\times \ell^2$)-continuous.*

See Appendix C.2 for details of the proof. As the target map $\Lambda^\star$ is a continuous and causal identifiable in-context mapping, by applying Lemma 4, $\Lambda^\star$ has the form (i), and is continuous on $\mathcal{X}_C^\sigma$. Also, the masked attention map $\bar{\Gamma}_\theta$ holds the same properties (See Lemma 11). Thus, it suffices to show that the following proposition.

**Proposition 2.** *For all $\varepsilon > 0$, there exist $L$ and parameters $(\theta_\ell, \xi_\ell)_{\ell=1}^L$ such that*

$$\forall (\mu_t, x) \in \mathcal{X}_C^\sigma, \ |F_{\xi_L} \diamond \bar{\Gamma}_{\theta_L} \diamond \ldots \diamond F_{\xi_1} \diamond \bar{\Gamma}_{\theta_1}(\mu_t, x) - \bar{\Lambda}^\star(\mu_t, x)| \leq \varepsilon,$$

*with $d_{\mathrm{tok}}(\theta_\ell) \leq d + 3d'$, $d_{\mathrm{head}}(\theta_\ell) = k(\theta_\ell) = 1$, $H(\theta_\ell) \leq d'$.*

The proof of Proposition 2 is basically the same as in the unmasked case, just replacing the arguments for $\mu \in \mathcal{P}(\Omega)$ with that for $\mu_t \in \mathrm{Lip}_C^\sigma(\tilde{\Omega})$. For the application of the Stone-Weierstrass theorem, we need to check the compactness of $\mathcal{X}_C^\sigma$, which is not obvious. Thus, we show that following lemma.

**Lemma 5.** *$\mathcal{X}_C^\sigma$ is compact for the (weak$^*$ $\times \ell^2$) topology.*

*Sketch of proof.* Assume that $\mu_n \in \mathrm{Lip}_C^\sigma(\tilde{\Omega})$ and $(x_n, t_n) \in \tilde{\Omega}$. We denote by $\mathcal{P}_\sigma([0,1]) := \{\nu \in \mathcal{P}([0,1]) : \nu(\{0\}) \geq \sigma\}$, and $\bar{\mu}_n \in \mathcal{P}_\sigma([0,1])$. As $\mathcal{P}_\sigma([0,1])$ and $\tilde{\Omega}$ are compact, there exits $\bar{\mu} \in \mathcal{P}_\sigma([0,1])$ and $(x, t) \in \tilde{\Omega}$ such that (if necessary, re-choosing a subsequence) $\bar{\mu}_n \rightharpoonup^* \bar{\mu}$ and $(x_n, t_n) \to (x, t)$. Since $[0,1] \ni s \mapsto \mu_n(\cdot|s) \in \mathcal{P}(\Omega)$ is $C$-Lipschitz, applying the general Arzelà–Ascoli theorem (see e.g., Kelley (2017, Chapter 7, Theorem 17)), there exists $\mu(\cdot|s) \in \mathcal{P}(\Omega)$ such that (if necessary, re-choosing a subsequence)

$$\sup_{s \in [0,1]} W_2(\mu_n(\cdot|s), \mu(\cdot|s)) \to 0 \text{ as } n \to \infty.$$

Thus, we define by $\mu := \mu(\cdot|s)\bar{\mu}(s)$ which belongs to $\mathrm{Lip}_C^\sigma(\tilde{\Omega})$. Using this convergence and the fact that the start points of $\bar{\mu}_n, \bar{\mu} \in \mathcal{P}_\sigma([0,1])$ are always fixed at $t = 0$, we can prove the convergence of the masked probability measures, i.e.,

$$(\mu_n)_{t_n} \rightharpoonup^* \mu_t \text{ as } n \to \infty.$$

The non-obvious situation is when $t_n > 0$ and $t = 0$ since the masked probability measure $\mu_t$ is differently defined on $t = 0$ or $t \in (0, 1]$. However, this can be solved by the continuity of the map $[0,1] \ni t \mapsto \mu_t \in \mathrm{Lip}_C^\sigma(\tilde{\Omega})$ (see Lemma 10). See Appendix C.3 for more details of the proof. □

## CONCLUSION AND DISCUSSION

In this work, we have presented a unified analysis of the expressivity of both unmasked and masked transformers in settings with an arbitrarily large number of tokens. A limitation of our method is that it is not quantitative. Using, for instance, the Wasserstein distance between token distributions could be a way to impose smoothness on the map to obtain quantitative bounds. Our proof relies on the approximation of the map along each dimension and the use of a commuting architecture (the transformer layers are multiplied together to obtain the output). This results in a growth of the number of heads proportional to the dimension. Lowering this dependency would require the development of new proof techniques beyond the use of the Stone-Weierstrass theorem. It is important to note that universality results like ours do not directly translate into conclusions about the learning capabilities of transformers. However, our proof techniques, which leverage ideas from optimal transport, share similarities with those used in the analysis of two-layer MLPs (Chizat & Bach, 2018). Thus, we believe that future work could build on this approach to investigate both convergence properties of the training of transformers. We also leave extending our approach to more recent positional encodings, such as Rotary Positional Embedding (RoPE) (Kazemnejad et al., 2024), for future work. RoPE modifies all attention layers to account for relative positional information, which would require slight adjustments to the proof to accommodate the different formulas.

## ACKNOWLEDGEMENTS

T. Furuya was supported by JSPS KAKENHI Grant Number JP24K16949, JST CREST JP-MJCR24Q5, and JST ASPIRE JPMJAP2329. M.V. de Hoop carried out the work while he was an invited professor at the Centre Sciences des Données at Ecole Normale Supérieure, Paris. He acknowledges the support of the Department of Energy, BES under grant DE-SC0020345, the Simons Foundation under the MATH + X Program, Oxy, and the corporate members of the Geo-Mathematical Imaging Group at Rice University. The work of G. Peyré was supported by the European Research Council (ERC project WOLF) and the French government under the management of Agence Nationale de la Recherche as part of the "France 2030" program, reference ANR-23-IACL-0008 (PRAIRIE-PSAI).

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

## APPENDIX

## A  BASIC NOTIONS

In this section, we briefly review basic notations used in the main text.

Let $\Omega \subset \mathbb{R}^d$ be a compact domain. For $T : \Omega \subset \mathbb{R}^d \to \Omega' \subset \mathbb{R}^{d'}$ being a measurable map, the push-forward $T_\sharp \mu \in \mathcal{P}(\Omega')$ of $\mu \in \mathcal{P}(\Omega)$ is given by

$$\forall A \subset \Omega', \quad T_\sharp \mu(A) := T\left(f^{-1}(A)\right).$$

The push-forward operator, $T_\sharp$, operates on discrete measures by simply displacing their supports,

$$T_\sharp \left(\frac{1}{n} \sum_{i=1}^n \delta_{x_i}\right) := \frac{1}{n} \sum_{i=1}^n \delta_{T(x_i)}.$$

For a general measure, $\nu = T_\sharp \mu$ is defined by a change of variables in integration, i.e.,

$$\forall g \in \mathcal{C}(\Omega'), \quad \int_{\Omega'} g(y) \, \mathrm{d}\nu(y) := \int_\Omega g(T(x)) \, \mathrm{d}\mu(x). \tag{22}$$

We employ the weak* topology on $\mathcal{P}(\Omega)$. This induces the following notion of convergence of sequences:

$$\mu_k \rightharpoonup^* \mu \quad \Leftrightarrow \quad \left(\forall f \in \mathcal{C}(\Omega), \int f(x) \, \mathrm{d}\mu_k(x) \to \int f(x) \, \mathrm{d}\mu(x)\right).$$

Intuitively, this corresponds to a "soft" notion of convergence where the support of $\mu_k$ approaches that of $\mu$.

In the special case of discrete measures, with a fixed number $n$ of points, this corresponds, up to relabeling of the points, to the usual convergence of points in finite dimensions:

$$\left(\frac{1}{n} \sum_{i=1}^n \delta_{x_i^k} \rightharpoonup^* \frac{1}{n} \sum_{i=1}^n \delta_{x_i}\right) \quad \Leftrightarrow \quad \left(X_k = (x_{i,k})_i \in \mathbb{R}^{d \times n} \to X = (x_i)_i \in \mathbb{R}^{d \times n}\right).$$

It is possible to metrize this weak* topology using the Wasserstein Optimal Transport distance, which is defined, for $1 \le p < +\infty$, as

$$W_p(\mu, \nu)^p := \min_{\pi \in \mathcal{P}(\Omega^2)} \left\{ \int \|x - y\|^p \, \mathrm{d}\pi(x, y) \ : \ \pi_1 = \mu, \pi_2 = \nu \right\},$$

where $\pi_i = (P_i)_\sharp \pi$ are the marginals of $\pi$ with $P_1(x, y) = x$ and $P_2(x, y) = y$. One has

$$\mu_k \rightharpoonup^* \mu \quad \Leftrightarrow \quad W_p(\mu_k, \mu) \to 0,$$

see e.g., Santambrogio (2015, Theorem 5.10). This Wasserstein distance is used in the masked case to impose a Lipschitz regularity with respect to the time of the contexts.

## B  PROOFS IN SECTION 3

### B.1  PROOF OF LEMMA 1

First, we show the one dimensional case of Lemma 1.

**Lemma 6.** *Let $\Omega \subset \mathbb{R}$ be a compact set, and let $\mu, \nu \in \mathcal{P}(\Omega)$. Then,*

$$L_1(\mu)(c) = L_1(\nu)(c), \ \forall c \in \mathbb{R}, \ \Rightarrow \ \mu = \nu.$$

*where, for $k \in \mathbb{N}$,*

$$L_k(\mu)(c) := \frac{\int e^{cy} y^k \, \mathrm{d}\mu(y)}{\int e^{cz} \, \mathrm{d}\mu(z)}.$$

*Proof.* One has
$$L_k(\mu)'(c) = L_{k+1}(\mu)(c) - L_k(\mu)(c)L_1(\mu)(c).$$
Hence, by recursion, we have that
$$L_1(\mu)(c) = L_1(\nu)(c), \ \forall c \in \mathbb{R}, \quad \Rightarrow \quad L_k(\mu)(c) = L_k(\nu)(c), \ \forall c \in \mathbb{R}, \ \forall k \geq 1,$$
Evaluating the equation at $c = 0$, we obtain that
$$L_k(\mu)(0) = L_k(\nu)(0), \ \forall k \geq 1 \quad \Leftrightarrow \quad \forall k, \int y^k \, \mathrm{d}\mu(y) = \int y^k \, \mathrm{d}\nu(y),$$
which is equivalent to $\mu = \nu$. $\qquad\square$

Using Lemma 6, we arrive at the following lemma.

**Lemma 7** (Lemma 1 in the main text). *Let $\Omega \subset \mathbb{R}^d$ be a compact set, and let $\mu, \nu \in \mathcal{P}(\Omega)$. Then,*
$$L(\mu)(a, c) = L(\nu)(a, c), \ \forall a \in \mathbb{R}^d, \forall c \in \mathbb{R}, \ \Rightarrow \ \mu = \nu.$$
*where*
$$L(\mu)(a, c) := \frac{\int \exp(c\langle a, \ y\rangle)\langle a, \ y\rangle \, \mathrm{d}\mu(y)}{\int \exp(c\langle a, \ z\rangle) \, \mathrm{d}\mu(z)}.$$

*Proof.* We define
$$\forall e \in \mathbb{S}^d, \quad \mu^e := (P_e)_\sharp \mu,$$
where $\mathbb{S}^d$ is the $d$-dimensional sphere, and $P_e(x) = \langle x, e\rangle$ is the projection on $e$. We see that
$$L(\mu)(e, c) = \frac{\int \exp(c\langle e, \ y\rangle)\langle e, \ y\rangle \, \mathrm{d}\mu(y)}{\int \exp(c\langle e, \ z\rangle) \, \mathrm{d}\mu(z)} = \frac{\int e^{cs} s \, \mathrm{d}\mu^e(s)}{\int e^{cr} \, \mathrm{d}\mu^e(r)}.$$
By Lemma 6, we can show that
$$\forall e, (P_e)_\sharp \mu = (P_e)_\sharp \nu,$$
which implies that, by the injectivity of the Radon transform (see e.g., Boman & Lindskog (2009, Theorem A)),
$$\mu = \nu.$$
$\qquad\square$

### B.2 Proof of Lemma 2

We write
$$\Lambda^\star(\mu, x) = (\Lambda_1^\star(\mu, x), \cdots, \Lambda_{d'}^\star(\mu, x)),$$
where $\Lambda_h^\star : \mathcal{P}(\Omega) \times \Omega \to \mathbb{R}$, so that
$$\Lambda^\star(\mu, x) = \sum_{h=1}^{d'} \Lambda_h^\star(\mu, x)e_h,$$
where $(e_h)_{h \in [d']}$ is the standard basis in $\mathbb{R}^{d'}$. For each $h = 1, \ldots, d'$, we apply Proposition 1 to $\Lambda_h^\star$ and conclude that there exist $T, N \in \mathbb{N}$ and $(\lambda_{t,n}^h)_{t \in [T], n \in [N]}$ such that
$$\forall(\mu, x) \in \mathcal{P}(\Omega) \times \Omega, \quad \left| \Lambda_h^\star(\mu, x) - \sum_{n=1}^N \gamma_{\lambda_{1,n}^h}(\mu, x) \odot \cdots \odot \gamma_{\lambda_{T,n}^h}(\mu, x) \right| \leq \frac{\varepsilon}{\sqrt{d'}}.$$
This implies that
$$\forall(\mu, x) \in \mathcal{P}(\Omega) \times \Omega, \quad \left| \Lambda^\star(\mu, x) - \sum_{h=1}^{d'} \left[ \sum_{n=1}^N \gamma_{\lambda_{1,n}^h}(\mu, x) \odot \cdots \odot \gamma_{\lambda_{T,n}^h}(\mu, x) \right] e_h \right|^2$$
$$\leq \sum_{h=1}^{d'} \left| \Lambda_h^\star(\mu, x) - \sum_{n=1}^N \gamma_{\lambda_{1,n}^h}(\mu, x) \odot \cdots \odot \gamma_{\lambda_{T,n}^h}(\mu, x) \right|^2 \leq \varepsilon^2. \quad (23)$$

Here, we wrote

$$\lambda_{t,n}^h = (a_{t,n}^h, b_{t,n}^h, c_{t,n}^h, v_{t,n}^h) \in \mathbb{R}^d \times \mathbb{R} \times \mathbb{R} \times \mathbb{R}.$$

We introduce

$$G(\mu, x) := \sum_{h=1}^{d'} \left[ \sum_{n=1}^{N} \gamma_{\lambda_{1,n}^h}(\mu, x) \odot \cdots \odot \gamma_{\lambda_{T,n}^h}(\mu, x) \right] e_h,$$

or

$$G(\mu, x) = \sum_{n=1}^{N} \bar{\gamma}_{\lambda_{1,n}}(\mu, x) \odot \cdots \odot \bar{\gamma}_{\lambda_{T,n}}(\mu, x), \tag{24}$$

in which

$$\bar{\gamma}_{\lambda_{t,n}}(\mu, x) := \left( \gamma_{\lambda_{t,n}^1}(\mu, x), ..., \gamma_{\lambda_{t,n}^{d'}}(\mu, x) \right) \in \mathbb{R}^{d'}.$$

We define self-attentions by

$$\Gamma_{\tilde{\theta}_{t,n}}(\mu, x) := x + \sum_{h=1}^{d'} \tilde{W}_{t,n}^h \int \frac{\exp\left( \langle \tilde{Q}_{t,n}^h x, \tilde{K}_{t,n}^h y \rangle \right)}{\int \exp\left( \langle \tilde{Q}_{t,n}^h x, \tilde{K}_{t,n}^h z \rangle \right) \mathrm{d}\mu(z)} \tilde{V}_{t,n}^h y \, \mathrm{d}\mu(y), \quad x \in \mathbb{R}^{d'}, \tag{25}$$

where

$$\tilde{\theta}_{t,n} := (\tilde{W}_{t,n}^h, \tilde{V}_{t,n}^h, \tilde{Q}_{t,n}^h, \tilde{K}_{t,n}^h)_{h=1,...,d'} \subset \mathbb{R}^{d' \times 1} \times \mathbb{R}^{1 \times d'} \times \mathbb{R}^{1 \times d'} \times \mathbb{R}^{1 \times d'},$$

i.e., $d_{\text{tok}}(\tilde{\theta}_{t,n}) = d'$, $d_{\text{head}}(\tilde{\theta}_{t,n}) = k(\tilde{\theta}_{t,n}) = 1$ and

$$\tilde{W}_{t,n}^h = (0, ..., 0, \underbrace{1}_{h-\text{th}}, 0, ..., 0) = e_h,$$

$$\tilde{V}_{t,n}^h = (0, ..., 0, \underbrace{v_{t,n}^h}_{h-\text{th}}, 0..., 0), \ \tilde{Q}_{t,n}^h = (0, ..., 0, \underbrace{c_{t,n}^h}_{h-\text{th}}, 0..., 0), \ \tilde{K}_{t,n}^h = (0, ..., 0, \underbrace{1}_{h-\text{th}}, 0..., 0).$$

We define affine transforms, $F_{\tilde{\xi}_{t,n}} : \mathbb{R}^d \to \mathbb{R}^{d'}$, according to

$$F_{\tilde{\xi}_{t,n}}(x) := A_{t,n} x + b_{t,n}, \tag{26}$$

where $\tilde{\xi}_{t,n} = (A_{t,n}, b_{t,n}) \in \mathbb{R}^{d' \times d} \times \mathbb{R}^{d'}$ in which

$$A_{t,n} = (a_{t,n}^1, ..., a_{t,n}^{d'}), \quad b_{t,n} = (b_{t,n}^1, ..., b_{t,n}^{d'}).$$

Then we have the composition,

$$\Gamma_{\tilde{\theta}_{t,n}} \diamond F_{\tilde{\xi}_{t,n}}(\mu, x) = \Gamma_{\tilde{\theta}_{t,n}}((F_{\tilde{\xi}_{t,n}})_\sharp \mu, F_{\tilde{\xi}_{t,n}}(x)) = A_{t,n} x + b_{t,n} \tag{27}$$

$$+ \sum_{h=1}^{d'} \tilde{W}_{t,n}^h \int \frac{\exp\left( \langle \tilde{Q}_{t,n}^h (A_{t,n} x + b_{t,n}), \tilde{K}_{t,n}^h (A_{t,n} y + b_{t,n}) \rangle \right)}{\int \exp\left( \langle \tilde{Q}_{t,n}^h (A_{t,n} x + b_{t,n}), \tilde{K}_{t,n}^h (A_{t,n} z + b_{t,n}) \rangle \right) \mathrm{d}\mu(z)}$$

$$\times \tilde{V}_{t,n}^h (A_{t,n} y + b_{t,n}) \, \mathrm{d}\mu(y) = \sum_{h=1}^{d'} \left[ \langle a_{t,n}^h, x \rangle + b_{t,n}^h \right.$$

$$\left. + \int \frac{\exp\left( (\langle a_{t,n}^h, x \rangle + b_{t,n}^h) c_{t,n}^h (\langle a_{t,n}^h, y \rangle + b_{t,n}^h) \right)}{\int \exp\left( (\langle a_{t,n}^h, x \rangle + b_{t,n}^h) c_{t,n}^h (\langle a_{t,n}^h, z \rangle + b_{t,n}^h) \right) \mathrm{d}\mu(z)} v_{t,n}^h (\langle a_{t,n}^h, y \rangle + b_{t,n}^h) \, \mathrm{d}\mu(y) \right] e_h$$

$$= \sum_{h=1}^{d'} \gamma_{\lambda_{t,n}^h}(\mu, x) e_h = \bar{\gamma}_{\lambda_{t,n}}(\mu, x).$$

With (24), we find that

$$G(\mu, x) = \sum_{n=1}^{N} (\Gamma_{\tilde{\theta}_{1,n}} \diamond F_{\tilde{\xi}_{1,n}})(\mu, x) \odot \cdots \odot (\Gamma_{\tilde{\theta}_{T,n}} \diamond F_{\tilde{\xi}_{T,n}})(\mu, x). \tag{28}$$

Therefore, with the estimate in (23), we obtain the statement in Lemma 2. □

### B.3 PROOF OF LEMMA 3

We first establish the following lemma.

**Lemma 8.** *For any $\varepsilon > 0$, there exists an MLP $\Phi : \mathbb{R}^{2d'} \to \mathbb{R}^{d'}$ such that*

$$\forall (\mu, x) \in \mathcal{P}(\Omega) \times \Omega, \quad |G(\mu, x) - G_\Phi(\mu, x)| \leq \varepsilon,$$

*where* $\quad G_\Phi(\mu, x) := \sum_{n=1}^{N} \Phi\Big((\Gamma_{\tilde{\theta}_{T,n}} \diamond F_{\tilde{\xi}_{T,n}})(\mu, x), \Phi\Big((\Gamma_{\tilde{\theta}_{T-1,n}} \diamond F_{\tilde{\xi}_{T-1,n}})(\mu, x),$

$$\cdots \Phi\Big((\Gamma_{\tilde{\theta}_{2,n}} \diamond F_{\tilde{\xi}_{2,n}})(\mu, x), \Phi\Big((\Gamma_{\tilde{\theta}_{1,n}} \diamond F_{\tilde{\xi}_{1,n}})(\mu, x), \mathbf{1}_{d'}\Big)\Big)\Big)\Big).$$

*Proof.* We note that

$$(\Gamma_{\tilde{\theta}_{1,n}} \diamond F_{\tilde{\xi}_{1,n}})(\mu, x) \odot \cdots \odot (\Gamma_{\tilde{\theta}_{T,n}} \diamond F_{\tilde{\xi}_{T,n}})(\mu, x)$$

$$= (\Gamma_{\tilde{\theta}_{T,n}} \diamond F_{\tilde{\xi}_{T,n}})(\mu, x) \odot \Big((\Gamma_{\tilde{\theta}_{T-1,n}} \diamond F_{\tilde{\xi}_{T-1,n}})(\mu, x) \odot \cdots$$

$$\odot \Big((\Gamma_{\tilde{\theta}_{2,n}} \diamond F_{\tilde{\xi}_{2,n}})(\mu, x) \odot \Big(\Gamma_{\tilde{\theta}_{1,n}} \diamond F_{\tilde{\xi}_{1,n}})(\mu, x) \odot \mathbf{1}_{d'}\Big)\Big)\Big).$$

Because the component-wise multiplication map $(x, y) \in \mathbb{R}^{2d'} \mapsto x \odot y \in \mathbb{R}^{d'}$ is continuous, by the universality of MLPs, for any $\varepsilon > 0$ and $R > 0$, there exists an MLP $\Phi : \mathbb{R}^{2d'} \to \mathbb{R}^{d'}$, such that

$$\forall (x, y) \in B_{\mathbb{R}^{2d'}}(0, R), \quad |x \odot y - \Phi(x, y)| \leq \varepsilon. \tag{29}$$

Since $\Omega \subset \mathbb{R}^d$ is compact then $0 \leq C_\Omega := \sup_{x \in \Omega} \|x\|_2$ is finite. Thus, using (27), we obtain the estimate,

$$\left|(\Gamma_{\tilde{\theta}_{t,n}} \diamond F_{\tilde{\xi}_{t,n}})(\mu, x)\right| \leq \|A_{t,n}\|_2 C_\Omega + \|b_{t,n}\|_2 + \sum_{h=1}^{d'} (\|A_{t,n}\|_2 C_\Omega + \|b_{t,n}\|_2) \|\tilde{W}_{t,n}^h \tilde{V}_{t,n}^h\|_2$$

$$\leq \max_{t \in [T], n \in [N]} \left((\|A_{t,n}\|_2 C_\Omega + \|b_{t,n}\|_2)(1 + \sum_{h=1}^{d'} \|\tilde{W}_{t,n}^h \tilde{V}_{t,n}^h\|_2)\right)$$

$$=: C_{\tilde{\Gamma}} \text{ for all } (\mu, x) \in \mathcal{P}(\Omega) \times \Omega, \quad (30)$$

where the constant, $C_{\tilde{\Gamma}} > 0$, depends on $\Omega$, $\tilde{W}_{t,n}^h$, $\tilde{V}_{t,n}^h$, $A_{t,n}$, $b_{t,n}$, but is independent of $t, n, \mu$ and $x$. Thus, using the universality in (29), choosing a large radius $R > 0$ depending on the constant $C_{\tilde{\Gamma}} > 0$, we can show that

$$\left| (\Gamma_{\tilde{\theta}_{1,n}} \diamond F_{\tilde{\xi}_{1,n}})(\mu, x) \odot \cdots \odot (\Gamma_{\tilde{\theta}_{T,n}} \diamond F_{\tilde{\xi}_{T,n}})(\mu, x) \right.$$

$$- \Phi\Big((\Gamma_{\tilde{\theta}_{T,n}} \diamond F_{\tilde{\xi}_{T,n}})(\mu, x), \Phi\Big((\Gamma_{\tilde{\theta}_{T-1,n}} \diamond F_{\tilde{\xi}_{T-1,n}})(\mu, x),$$

$$\left. \cdots \Phi\Big((\Gamma_{\tilde{\theta}_{2,n}} \diamond F_{\tilde{\xi}_{2,n}})(\mu, x), \Phi\Big((\Gamma_{\tilde{\theta}_{1,n}} \diamond F_{\tilde{\xi}_{1,n}})(\mu, x), \mathbf{1}_{d'}\Big)\Big)\Big)\Big) \right| \leq \frac{\varepsilon}{N}.$$

Upon summing from $n = 1, \ldots, N$ and using the form (17), we complete the proof of Lemma 8. $\quad\square$

**Remark 3.** *(The challenge to derive quantitative estimates.) The key is to approximate and capture the mentioned multiplicity by MLPs, for which quantitative estimates have been studied, e.g., Elbrächter et al. (2022, Lemma 6.2), which is a variant of Yarotsky (2017, Proposition 2). However, the depth and width of MLPs depend on the bound of input variables. Specifically, an existential $\Phi$ in the above depends on the bound $C_{\tilde{\Gamma}}$ (see (30)), which in turn depends on parameters in $\Gamma_{\tilde{\theta}_{t,n}} \diamond F_{\tilde{\xi}_{t,n}}$ that are chosen to approximate $\Gamma^*$ within $\varepsilon$ through the application of the Stone-Weierstrass theorem (see Proposition 1). Thus, providing the quantitative estimate for the MLP, $\Phi$, is challenging.*

Finally in this appendix we prove, by construction, the following result.

**Lemma 9.** *Let* $\Phi : \mathbb{R}^{2d'} \to \mathbb{R}^{d'}$ *be an MLP. There exist* $\xi_0$, $(\xi_{t,n})_{t \in [T], n \in [N]}$, $\xi_*$, *and* $\theta_0$, $(\theta_{t,n})_{t \in [T], n \in [N]}$, $\theta_*$ *such that*

$$\forall (\mu, x) \in \mathcal{P}(\Omega) \times \Omega, \quad G_\Phi(\mu, x) = F_{\xi_*} \diamond \Gamma_{\theta_*} \diamond \left( \diamond_{n=1}^N \diamond_{t=1}^T F_{\xi_{t,n}} \diamond \Gamma_{\theta_{t,n}} \right) \diamond F_{\xi_0} \diamond \Gamma_{\theta_0}(\mu, x).$$

*with following sizes:*

$$d_{\text{tok}}(\theta_0) = d, \quad d_{\text{head}}(\theta_0) = k(\theta_0) = H(\theta_0) = 1,$$
$$d_{\text{tok}}(\theta_{t,n}) = d + 3d', \quad d_{\text{head}}(\theta_{t,n}) = k(\theta_{t,n}) = 1, \quad H(\theta_{t,n}) = d',$$
$$d_{\text{tok}}(\theta_*) = d + 3d', \quad d_{\text{head}}(\theta_*) = k(\theta_*) = H(\theta_*) = 1.$$

*Proof.* The proof is based on the following scheme:

$$x \xrightarrow[\text{[Step A]}]{F_{\xi_0} \diamond \Gamma_{\theta_0}} \begin{pmatrix} x \\ F_{\tilde{\xi}_{1,1}}(x) \\ \varphi_{1,1}(x) \\ f_1(x) \end{pmatrix}$$

$$\xrightarrow[\text{[Step B]}]{F_{\xi_{1,1}} \diamond \Gamma_{\theta_{1,1}}} \begin{pmatrix} x \\ F_{\tilde{\xi}_{2,1}}(x) \\ \varphi_{2,1}(x) \\ f_1(x) \end{pmatrix} \xrightarrow[\text{[Step B]}]{F_{\xi_{2,1}} \diamond \Gamma_{\theta_{2,1}}} \cdots \xrightarrow[\text{[Step B]}]{F_{\xi_{T-1,1}} \diamond \Gamma_{\theta_{T-1,1}}} \begin{pmatrix} x \\ F_{\tilde{\xi}_{T,1}}(x) \\ \varphi_{T,1}(x) \\ f_1(x) \end{pmatrix} \xrightarrow[\text{[Step C]}]{F_{\xi_{T,1}} \diamond \Gamma_{\theta_{T,1}}} \begin{pmatrix} x \\ F_{\tilde{\xi}_{2,1}}(x) \\ \varphi_{1,2}(x) \\ f_2(x) \end{pmatrix}$$

$$\xrightarrow[\text{[Step B]}]{F_{\xi_{1,2}} \diamond \Gamma_{\theta_{1,2}}} \begin{pmatrix} x \\ F_{\tilde{\xi}_{2,2}}(x) \\ \varphi_{2,2}(x) \\ f_2(x) \end{pmatrix} \xrightarrow[\text{[Step B]}]{F_{\xi_{2,2}} \diamond \Gamma_{\theta_{2,2}}} \cdots \xrightarrow[\text{[Step B]}]{F_{\xi_{T-1,2}} \diamond \Gamma_{\theta_{T-1,2}}} \begin{pmatrix} x \\ F_{\tilde{\xi}_{T,2}}(x) \\ \varphi_{T,2}(x) \\ f_2(x) \end{pmatrix} \xrightarrow[\text{[Step C]}]{F_{\xi_{T,2}} \diamond \Gamma_{\theta_{T,2}}} \begin{pmatrix} x \\ F_{\tilde{\xi}_{1,3}}(x) \\ \varphi_{1,3}(x) \\ f_3(x) \end{pmatrix}$$

$$\vdots$$

$$\xrightarrow[\text{[Step B]}]{F_{\xi_{1,N}} \diamond \Gamma_{\theta_{1,N}}} \begin{pmatrix} x \\ F_{\tilde{\xi}_{2,N}}(x) \\ \varphi_{2,N}(x) \\ f_N(x) \end{pmatrix} \xrightarrow[\text{[Step B]}]{F_{\xi_{2,N}} \diamond \Gamma_{\theta_{2,N}}} \cdots \xrightarrow[\text{[Step B]}]{F_{\xi_{T-1,N}} \diamond \Gamma_{\theta_{T-1,N}}} \begin{pmatrix} x \\ F_{\tilde{\xi}_{T,N}}(x) \\ \varphi_{T,N}(x) \\ f_N(x) \end{pmatrix} \xrightarrow[\text{[Step C]}]{F_{\xi_{T,N}} \diamond \Gamma_{\theta_{T,N}}} \begin{pmatrix} x \\ F_{\tilde{\xi}_{1,N+1}}(x) \\ \varphi_{1,N+1}(x) \\ f_{N+1}(x) = \end{pmatrix}$$

$$\xrightarrow[\text{[Step D]}]{F_{\xi_*} \diamond \Gamma_{\theta_*}} f_{N+1}(x) = G_\Phi(\mu, x)$$

where $\varphi_{t,n} : \mathbb{R}^d \to \mathbb{R}^{d'}$ is given by

$$\varphi_{t,n}(x) := \begin{cases} \Phi\Big( (\Gamma_{\tilde{\theta}_{t,n}} \diamond F_{\tilde{\xi}_{t,n}})(\mu, x), \Phi\Big( (\Gamma_{\tilde{\theta}_{t-1,n}} \diamond F_{\tilde{\xi}_{t-1,n}})(\mu, x) \\ \cdots \Phi\Big( (\Gamma_{\tilde{\theta}_{2,n}} \diamond F_{\tilde{\xi}_{2,n}})(\mu, x), \Phi\Big( (\Gamma_{\tilde{\theta}_{1,n}} \diamond F_{\tilde{\xi}_{1,n}})(\mu, x), \mathbf{1}_{d'} \Big) \Big) \Big) \Big) \Big), & t \geq 2 \\ \mathbf{1}_{d'}, & t = 1 \end{cases},$$

and $f_n : \mathbb{R}^d \to \mathbb{R}^{d'}$ by

$$f_n(x) := \begin{cases} \sum_{i=1}^{n-1} \varphi_{T,i}(x), & n \geq 2 \\ 0 & n = 1 \end{cases},$$

where $\Gamma_{\tilde{\theta}_{t,n}}$ and $F_{\tilde{\xi}_{t,n}} : \mathbb{R}^d \to \mathbb{R}^{d'}$ are the self-attention and affine maps chosen in (25) and (26), respectively. Here, $\Gamma_{\theta_0}$, $\Gamma_{\theta_{t,n}}$, $\Gamma_{\theta_*}$, $F_{\xi_0}$, $F_{\xi_{t,n}}$ and $F_{\xi_*}$ will be specified below, in the following steps:

**[Step A]** Let $\Gamma_{\theta_0}(\mu) : \mathbb{R}^d \to \mathbb{R}^d$ be

$$\Gamma_{\theta_0}(\mu, x) = x,$$

and let $F_{\xi_0} : \mathbb{R}^d \to \mathbb{R}^{d+3d'}$ be the affine transform defined by

$$F_{\xi_0}(x) := (x, A_{1,1}x + b_{1,1}, \mathbf{1}_{d'}, 0) = (x, F_{\tilde{\xi}_{1,1}}(x), \varphi_{1,1}(x), f_1(x)).$$

Then we see that

$$F_{\xi_0} \diamond \Gamma_{\theta_0}(\mu, x) = (x, F_{\tilde{\xi}_{1,1}}(x), \varphi_{1,1}(x), f_1(x)),$$

and

$$\mu_{1,1} := (F_{\xi_0} \diamond \Gamma_{\theta_0}(\mu))_\sharp \mu = \Big( \mu, (F_{\tilde{\xi}_{1,1}})_\sharp \mu, (\varphi_{1,1})_\sharp \mu, (f_1)_\sharp \mu \Big).$$

We proceed with **[Step B]** in which we handle the case when $n = t = 1$.

**[Step B]** Let $t = 1, ..., T - 1$ and $n = 1, ..., N$. We already have that

$$\left(\diamond_{j=1}^{t-1} F_{\xi_{j,n}} \diamond \Gamma_{\theta_{j,n}}\right) \diamond \left(\diamond_{i=1}^{n-1} \diamond_{s=1}^{T} F_{\xi_{s,i}} \diamond \Gamma_{\theta_{s,i}}\right) \diamond F_{\xi_0} \diamond \Gamma_{\theta_0}(\mu, x) = \left(x, F_{\tilde{\xi}_{t,n}}(x), \varphi_{t,n}(x), f_n(x)\right),$$

and

$$\mu_{t,n} := \left(\left(\diamond_{j=1}^{t-1} F_{\xi_{j,n}} \diamond \Gamma_{\theta_{j,n}}\right) \diamond \left(\diamond_{i=1}^{n-1} \diamond_{s=1}^{T} F_{\xi_{s,i}} \diamond \Gamma_{\theta_{s,i}}\right) \diamond F_{\xi_0} \diamond \Gamma_{\theta_0}(\mu)\right)_{\sharp} \mu$$

$$= \left(\mu, (F_{\tilde{\xi}_{t,n}})_{\sharp}\mu, (\varphi_{t,n})_{\sharp}\mu, (f_n)_{\sharp}\mu\right).$$

When $n = 1$ or $t = 1$, the above reduces to $\diamond_{i=1}^{n-1} \diamond_{s=1}^{T} F_{\xi_{s,i}} \diamond \Gamma_{\theta_{s,i}} = I_{d+3d'}$ or $\diamond_{j=1}^{t-1} F_{\xi_{j,n}} \diamond \Gamma_{\theta_{j,n}} = I_{d+3d'}$.

Let $\Gamma_{\theta_{t,n}}(\mu_{t,n}) : \mathbb{R}^{d+3d'} \to \mathbb{R}^{d+3d'}$ be given by

$$\Gamma_{\theta_{t,n}}(\mu_{t,n}, (x, u, p, w)) = (x, u, p, w)$$

$$+ \sum_{h=1}^{d'} W_{t,n}^h \int \frac{\exp\left(\langle Q_{t,n}^h(x, u, p, w), K_{t,n}^h(y', v', q', z')\rangle\right)}{\int \exp\left(\langle Q_{t,n}^h(x, u, p, w), K_{t,n}^h(y, v, q, z)\rangle\right) \mathrm{d}\mu_{t,n}(y, v, q, z)}$$

$$V_{t,n}^h(y', v', q', z') \, \mathrm{d}\mu_{t,n}(y', v', q', z')$$

$$= \left(x, u + \sum_{h=1}^{d'} \tilde{W}_{t,n}^h \int \frac{\exp\left(\langle \tilde{Q}_{t,n}^h u, \tilde{K}_{t,n}^h v'\rangle\right)}{\int \exp\left(\langle \tilde{Q}_{t,n}^h u, \tilde{K}_{t,n}^h v\rangle\right) \mathrm{d}\mu_{t,n}(y, v, q, z)} \tilde{V}_{t,n}^h v' \, \mathrm{d}\mu_{t,n}(y', v', q', z'), p, w\right)$$

$$= \left(x, \Gamma_{\tilde{\theta}_{t,n}}((F_{\tilde{\xi}_{t,n}})_{\sharp}\mu, u), p, w\right),$$

where $x, y, y' \in \mathbb{R}^d$, $u, v, v' \in \mathbb{R}^{d'}$, $p, q, q' \in \mathbb{R}^{d'}$, and $w, z, z' \in \mathbb{R}^{d'}$. Here, $\theta_{t,n}$ is given by

$$\theta_{t,n} := (W_{t,n}^h, V_{t,n}^h, Q_{t,n}^h, K_{t,n}^h)_{h=1,...,d'} \subset \mathbb{R}^{d+3d' \times 1} \times \mathbb{R}^{1 \times d+3d'} \times \mathbb{R}^{1 \times d+3d'} \times \mathbb{R}^{1 \times d+3d'},$$

that is,

$$d_{\mathrm{tok}}(\theta_{t,n}) = d + 3d', \quad d_{\mathrm{head}}(\theta_{t,n}) = k(\theta_{t,n}) = 1, \quad H(\theta_{t,n}) = d',$$

and

$$W_{t,n}^h := (O, \tilde{W}_{t,n}^h, O, O), \ V_{t,n}^h := (O, \tilde{V}_{t,n}^h, O, O),$$

$$Q_{t,n}^h := (O, \tilde{Q}_{t,n}^h, O, O), \ K_{t,n}^h := (O, \tilde{K}_{t,n}^h, O, O).$$

Let $F_{\xi_{t,n}} : \mathbb{R}^{d+3d'} \to \mathbb{R}^{d+3d'}$ be defined by

$$F_{\xi_{t,n}}(x, u, p, w) = (x, A_{t+1,n}x + b_{t+1,n}, \Phi(u, p), w) = (x, F_{\tilde{\xi}_{t+1,n}}(x), \Phi(u, p), w). \tag{31}$$

Then we have

$$\left(\diamond_{j=1}^{t} F_{\xi_{j,n}} \diamond \Gamma_{\theta_{j,n}}\right) \diamond \left(\diamond_{i=1}^{n-1} \diamond_{s=1}^{T} F_{\xi_{s,i}} \diamond \Gamma_{\theta_{s,i}}\right) \diamond F_{\xi_0} \diamond \Gamma_{\theta_0}(\mu, x)$$

$$= F_{\xi_{t,n}} \diamond \Gamma_{\theta_{t,n}} \diamond \left(\diamond_{j=1}^{t-1} F_{\xi_{j,n}} \diamond \Gamma_{\theta_{j,n}}\right) \diamond \left(\diamond_{i=1}^{n-1} \diamond_{s=1}^{T} F_{\xi_{s,i}} \diamond \Gamma_{\theta_{s,i}}\right) \diamond F_{\xi_0} \diamond \Gamma_{\theta_0}(\mu, x)$$

$$= F_{\xi_{t,n}} \diamond \Gamma_{\theta_{t,n}}(\mu_{t,n}, (x, F_{\tilde{\xi}_{t,n}}(x), \varphi_{t,n}(x), f_n(x)))$$

$$= F_{\xi_{t,n}}(x, \Gamma_{\tilde{\theta}_{t,n}}((F_{\tilde{\xi}_{t,n}})_{\sharp}\mu, F_{\tilde{\xi}_{t,n}}(x)), \varphi_{t,n}(x), f_n(x))$$

$$= F_{\xi_{t,n}}(x, (\Gamma_{\tilde{\theta}_{t,n}} \diamond F_{\tilde{\xi}_{t,n}})(\mu, x), \varphi_{t,n}(x), f_n(x))$$

$$= \left(x, F_{\tilde{\xi}_{t+1,n}}(x), \varphi_{t+1,n}(x), f_n(x)\right),$$

and

$$\mu_{t+1,n} := \left(\left(\diamond_{j=1}^{t} F_{\xi_{j,n}} \diamond \Gamma_{\theta_{j,n}}\right) \diamond \left(\diamond_{i=1}^{n-1} \diamond_{s=1}^{T} F_{\xi_{s,i}} \diamond \Gamma_{\theta_{s,i}}\right) \diamond F_{\xi_0} \diamond \Gamma_{\theta_0}(\mu)\right)_{\sharp} \mu$$

$$= \left(\mu, (F_{\tilde{\xi}_{t+1,n}})_{\sharp}\mu, (\varphi_{t+1,n})_{\sharp}\mu, (f_n)_{\sharp}\mu\right).$$

We repeat **[Step B]** until obtaining $\mu_{T,n}$. Once $\mu_{T,n}$ is obtained, we proceed with **[Step C]**.

**[Step C]** Let $\Gamma_{\theta_{T,n}}(\mu_{T,n}) : \mathbb{R}^{d+3d'} \to \mathbb{R}^{d+3d'}$ be given by

$$
\Gamma_{\theta_{T,n}}(\mu_{T,n}, (x, u, p, w))
$$

$$
= \left( x, u + \sum_{h=1}^{d'} \tilde{W}_{T,n}^h \int \frac{\exp\left( \langle \tilde{Q}_{T,n}^h u, \ \tilde{K}_{T,n}^h v' \rangle \right)}{\int \exp\left( \langle \tilde{Q}_{T,n}^h u, \ \tilde{K}_{T,n}^h v \rangle \right) \mathrm{d}\mu_{T,n}(y, v, q, z)} \tilde{V}_{T,n}^h v' \, \mathrm{d}\mu_{T,n}(y', v', q', z'), p, w \right)
$$

$$
= \left( x, \Gamma_{\tilde{\theta}_{T,n}}((F_{\tilde{\xi}_{T,n}})_\sharp \mu, u), p, w \right).
$$

Let $F_{\xi_{T,n}} : \mathbb{R}^{d+3d'} \to \mathbb{R}^{d+3d'}$ be defined by

$$
F_{\xi_{T,n}}(x, u, p, w) = (x, A_{1,n+1}x + b_{1,n+1}, \mathbf{1}_{d'}, w + \Phi(u, p)) = (x, F_{\tilde{\xi}_{1,n+1}}(x), \varphi_{1,n+1}(x), w + \Phi(u, p)). \tag{32}
$$

When $n = N$, we define by $F_{\tilde{\xi}_{1,N+1}}(x) := 0$ and $\varphi_{1,N+1} := 0$ in the above. We find that

$$
(\diamond_{j=1}^T F_{\xi_{j,n}} \diamond \Gamma_{\theta_{j,n}}) \diamond (\diamond_{i=1}^{n-1} \diamond_{s=1}^T F_{\xi_{s,i}} \diamond \Gamma_{\theta_{s,i}}) \diamond F_{\xi_0} \diamond \Gamma_{\theta_0}(\mu, x)
$$

$$
= F_{\xi_{T,n}}(x, (\Gamma_{\tilde{\theta}_{T,n}} \diamond F_{\tilde{\theta}_{T,n}})(\mu, x), \varphi_{T,n}(x), f_n(x))
$$

$$
= \left( x, F_{\tilde{\xi}_{1,n+1}}(x), \varphi_{1,n+1}(x), f_{n+1}(x) \right),
$$

and

$$
\mu_{T+1,n} := \left( \left( \diamond_{j=1}^T F_{\xi_{j,n}} \diamond \Gamma_{\theta_{j,n}} \right) \diamond \left( \diamond_{i=1}^{n-1} \diamond_{s=1}^T F_{\xi_{s,i}} \diamond \Gamma_{\theta_{s,i}} \right) \diamond F_{\xi_0} \diamond \Gamma_{\theta_0}(\mu) \right)_\sharp \mu
$$

$$
= \left( \mu, (F_{\tilde{\xi}_{1,n+1}})_\sharp \mu, (\varphi_{1,n+1})_\sharp \mu, (f_{n+1})_\sharp \mu \right).
$$

Denoting

$$
\mu_{1,n+1} := \mu_{T+1,n},
$$

we return to **[Step B]**, and repeat **[Step B]** and **[Step C]** until obtaining $\mu_{T+1,N}$. Once $\mu_{T+1,N}$ is obtained, we proceed with **[Step D]**.

**[Step D]** Let $\Gamma_{\theta_*}(\mu_{T+1,N}) : \mathbb{R}^{d+3d'} \to \mathbb{R}^{d+3d'}$ be given by

$$
\Gamma_{\theta_*}(\mu_{T+1,N}, (x, u, p, w)) = (x, u, p, w),
$$

and let $F_{\xi_*} : \mathbb{R}^{d+3d'} \to \mathbb{R}^{d'}$ be the affine transform defined by

$$
F_{\xi_*}(x, u, p, w) := w.
$$

Then we conclude that

$$
F_{\xi_*} \diamond \Gamma_{\theta_*} \diamond \left( \diamond_{n=1}^N \diamond_{s=1}^T F_{\xi_{s,n}} \diamond \Gamma_{\theta_{s,n}} \right) \diamond F_{\xi_0} \diamond \Gamma_{\theta_0}(\mu, x)
$$

$$
= F_{\xi_*} \diamond \Gamma_{\theta_*} \left( \mu_{T+1,N}, \left( x, F_{\tilde{\xi}_{1,N+1}}(x), \varphi_{1,n+1}(x), f_{N+1}(x) \right) \right) = f_{N+1}(x)
$$

$$
= \sum_{n=1}^N \Phi\Big( (\Gamma_{\tilde{\theta}_{T,n}} \diamond F_{\tilde{\xi}_{T,n}})(\mu, x), \Phi\Big( (\Gamma_{\tilde{\theta}_{T-1,n}} \diamond F_{\tilde{\xi}_{T-1,n}})(\mu, x),
$$

$$
\cdots \Phi\Big( (\Gamma_{\tilde{\theta}_{2,n}} \diamond F_{\tilde{\xi}_{2,n}})(\mu, x), \Phi\Big( (\Gamma_{\tilde{\theta}_{1,n}} \diamond F_{\tilde{\xi}_{1,n}})(\mu, x), \mathbf{1}_{d'} \Big) \Big) \Big) \Big) = G_\Phi(\mu, x).
$$

$\square$

**Remark 4.** *Note that if the MLPs represent the identity map, such as when using ReLU activation functions, then context-free maps $F_{\xi_{t,n}}$ in (31) and (32) can be represented by MLPs. If this is not the case, it is sufficient to further approximate $F_{\xi_{T,n}}$ using MLPs.*

## C   Proofs in Section 4

### C.1   Basic properties for the masked case

**Lemma 10.** *The map $[0,1] \ni t \mapsto \mu_t \in \mathrm{Lip}_C^\sigma(\tilde{\Omega})$ is continuous (for the weak* topology).*

*Proof.* Let $\mu \in \mathrm{Lip}_C^\sigma(\tilde{\Omega})$. We re-define the masked probability measure (including at $t = 0$) by

$$\mu_t := \begin{cases} \frac{1_{[0,t]}}{\bar{\mu}([0,t])} \cdot \mu & t \in (0,1] \\ \mu(\cdot|s)\delta_{s=0} & t = 0 \end{cases}. \tag{33}$$

Thus, the continuity on $t \in (0,1]$ is obvious. We now show that as $t \to 0$

$$\mu_t \rightharpoonup^* \mu_0.$$

For $f \in \mathcal{C}(\tilde{\Omega})$, we see that

$$\left| \int f(x,s) \, \mathrm{d}\mu_t - \int f(x,s) \, \mathrm{d}\mu_0 \right|$$

$$= \left| \int f(x,s) \, \mathrm{d}\mu(x|s) \frac{1_{[0,t]}(s)}{\bar{\mu}([0,t])} \, \mathrm{d}\bar{\mu}(s) - \int f(x,0) \, \mathrm{d}\mu(x|0) \right|$$

$$= \left| \int f(x,s) \, \mathrm{d}\mu(x|s) \frac{1_{[0,t]}(s)}{\bar{\mu}([0,t])} \, \mathrm{d}\bar{\mu}(s) - \int f(x,0) \, \mathrm{d}\mu(x|0) \, \mathrm{d} \frac{1_{[0,t]}(s)}{\bar{\mu}([0,t])} \, \mathrm{d}\bar{\mu}(s) \right|$$

$$\leq \int \left| \frac{1_{[0,t]}(s)}{\bar{\mu}([0,t])} (F(s) - F(0)) \right| \mathrm{d}\bar{\mu}(s) \leq \sup_{s \in [0,t]} |F(s) - F(0)|,$$

where

$$F(s) := \int f(x,s) \, \mathrm{d}\mu(x|s),$$

and $s \mapsto F(s)$ is continuous as $\mu \in \mathrm{Lip}_C^\sigma(\tilde{\Omega})$. Thus we have, as $t \to 0$,

$$\int f(x,s) \, \mathrm{d}\mu_t \to \int f(x,s) \, \mathrm{d}\mu_0,$$

which implies that

$$\mu_t \rightharpoonup^* \mu_0.$$

$\square$

**Lemma 11.** *Let $\Gamma_\theta$ be the masked in-context map defined in (12). Then we have the following:*

*(a)  $\Gamma_\theta$ is a causal identifiable in-context map in the sense of Definition 3.*

*(b)  For any $(\mu, x, t) \in \mathrm{Lip}_C^\sigma(\tilde{\Omega}) \times \tilde{\Omega}$,*
$$\Gamma_\theta(\mu, x, t) = \bar{\Gamma}_\theta(\mu_t, x).$$

*(c)  The reduced map of $\mathcal{X}_C^\sigma \ni (\mu_t, x) \mapsto \bar{\Gamma}_\theta(\mu_t, x)$ is (weak* $\times \ell^2$)-continuous.*

*Proof.* To show (a), we observe that

$$\Gamma_\theta(\mu, x, t) = x + \int \frac{\exp\left(\frac{1}{\sqrt{k}}\langle Q^h x, K^h y\rangle\right) 1_{[0,t]}(r)}{\int \exp\left(\frac{1}{\sqrt{k}}\langle Q^h x, K^h z\rangle\right) 1_{[0,t]}(s) \, \mathrm{d}\mu(z,s)} V^h y \, \mathrm{d}\mu(y,r)$$

$$= x + \int \frac{\exp\left(\frac{1}{\sqrt{k}}\langle Q^h x, K^h y\rangle\right)}{\int \exp\left(\frac{1}{\sqrt{k}}\langle Q^h x, K^h z\rangle\right) \frac{1_{[0,t]}(s)}{\bar{\mu}([0,t])} \, \mathrm{d}\mu(z,s)} V^h y \frac{1_{[0,t]}(r)}{\bar{\mu}([0,t])} \, \mathrm{d}\mu(y,r)$$

$$= x + \int \frac{\exp\left(\frac{1}{\sqrt{k}}\langle Q^h x, K^h y\rangle\right) 1_{[0,t]}(r)}{\int \exp\left(\frac{1}{\sqrt{k}}\langle Q^h x, K^h z\rangle\right) 1_{[0,t]}(s) \, \mathrm{d}\mu_t(z,s)} V^h y \, \mathrm{d}\mu_t(y,r) = \Gamma_\theta(\mu_t, x, t).$$

$$\tag{34}$$

This proves the causality. To show the identifiability, we assume that $\mu_t = \mu_{t'}$ where $\mu \in \mathrm{Lip}_C^\sigma(\tilde{\Omega})$ and $t, t' \in [0, 1]$. Without loss of generality, we assume that $t < t'$. Then we have that $\bar{\mu} = 0$ on $[t, t']$, so that

$$
\Gamma_\theta(\mu_t, x, t) = x + \int \frac{\exp\left(\frac{1}{\sqrt{k}}\langle Q^h x, K^h y\rangle\right) 1_{[0,t]}(r)}{\int \exp\left(\frac{1}{\sqrt{k}}\langle Q^h x, K^h z\rangle\right) 1_{[0,t]}(s)\,\mathrm{d}\mu_t(z, s)} V^h y \,\mathrm{d}\mu_t(y, r)
$$

$$
= x + \int \frac{\exp\left(\frac{1}{\sqrt{k}}\langle Q^h x, K^h y\rangle\right) 1_{[0,t]}(r)}{\int \exp\left(\frac{1}{\sqrt{k}}\langle Q^h x, K^h z\rangle\right) 1_{[0,t]}(s)\,\mathrm{d}\mu_{t'}(z, s)} V^h y \,\mathrm{d}\mu_{t'}(y, r)
$$

$$
= x + \int \frac{\exp\left(\frac{1}{\sqrt{k}}\langle Q^h x, K^h y\rangle\right) 1_{[0,t']}(r)}{\int \exp\left(\frac{1}{\sqrt{k}}\langle Q^h x, K^h z\rangle\right) 1_{[0,t']}(s)\,\mathrm{d}\mu_{t'}(z, s)} V^h y \,\mathrm{d}\mu_{t'}(y, r) = \Gamma_\theta(\mu_{t'}, x, t').
$$

Thus we obtain (a).

From Lemma 11 (a), $\Gamma_\theta$ is a causal identifiable in-context map in the sense of Definition 3. By applying Lemma 4 (i), as $\Lambda = \Gamma_\theta$, we obtain (b).

Using Lemma 11 (b), we find that

$$
\bar{\Gamma}_\theta(\mu_t, x) = \Gamma_\theta(\mu_t, x, t)
$$

$$
= x + \int \frac{\exp\left(\frac{1}{\sqrt{k}}\langle Q^h x, K^h y\rangle\right) 1_{[0,t]}(r)}{\int \exp\left(\frac{1}{\sqrt{k}}\langle Q^h x, K^h z\rangle\right) 1_{[0,t]}(s)\,\mathrm{d}\mu_t(z, s)} V^h y \,\mathrm{d}\mu_t(y, r)
$$

$$
= x + \int \frac{\exp\left(\frac{1}{\sqrt{k}}\langle Q^h x, K^h y\rangle\right)}{\int \exp\left(\frac{1}{\sqrt{k}}\langle Q^h x, K^h z\rangle\right)\,\mathrm{d}\mu_t(z, s)} V^h y \,\mathrm{d}\mu_t(y, r). \tag{35}
$$

We can show the continuity of the map

$$
\mathcal{P}(\tilde{\Omega}) \times \Omega \ni (\mu, x) \mapsto x + \int \frac{\exp\left(\frac{1}{\sqrt{k}}\langle Q^h x, K^h y\rangle\right)}{\int \exp\left(\frac{1}{\sqrt{k}}\langle Q^h x, K^h z\rangle\right)\,\mathrm{d}\mu(z, s)} V^h y \,\mathrm{d}\mu(y, r) \in \mathbb{R}^{d'},
$$

which, in fact, follows from the continuity of the unmasked self-attention. Thus, with (35), we obtain (c). $\qquad \square$

**Lemma 12.** *Let $\Gamma_1$ and $\Gamma_2$ be causal identifiable in-context maps in the sense of Definition 3. Then, the composition $\Gamma_2 \diamond \Gamma_1$ in the sense of* (13) *is a causal identifiable in-context map.*

*Proof.* Assume that $(\mu, x, t) \in \mathrm{Lip}_C^\sigma(\tilde{\Omega}) \times \tilde{\Omega}$. We first show that

$$
[(\Gamma_1(\mu), \mathrm{Id}_\mathbb{R})_\sharp \mu]_t = (\Gamma_1(\mu_t), \mathrm{Id}_\mathbb{R})_\sharp \mu_t. \tag{36}
$$

Indeed, we see that for all $f \in \mathcal{C}(\tilde{\Omega})$

$$
\int f(x, s)\,\mathrm{d}\left[(\Gamma_1(\mu), \mathrm{Id}_\mathbb{R})_\sharp \mu\right](x, s) = \int f\left(\Gamma_1(\mu)(x, s), s\right)\,\mathrm{d}\mu(x, s)
$$

$$
= \int f\left(\Gamma_1(\mu)(x, s), s\right)\,\mathrm{d}\mu(x|s)\,\mathrm{d}\bar{\mu}(s)
$$

$$
= \int f(x, s)\,\mathrm{d}\left[\Gamma_1(\mu)(\cdot, s)_\sharp \mu(\cdot|s)\right](x)\,\mathrm{d}\bar{\mu}(s),
$$

which obtains that

$$
(\Gamma_1(\mu), \mathrm{Id}_\mathbb{R})_\sharp \mu = \left[\Gamma_1(\mu)(\cdot, s)_\sharp \mu(\cdot|s)\right] \bar{\mu}(s). \tag{37}
$$

This implies that by using the causality of $\Gamma_1$

$$\int f(x, s) \, \mathrm{d} \left[ (\Gamma_1(\mu), \mathrm{Id}_{\mathbb{R}})_\sharp \mu \right]_t (x, s)$$

$$= \int f(x, s) \, \mathrm{d} \left[ \Gamma_1(\mu)(\cdot, s)_\sharp \mu(\cdot|s) \right](x) \frac{1_{[0,t]}(s)}{\bar{\mu}([0,t])} \, \mathrm{d}\bar{\mu}(s)$$

$$= \int f(\Gamma_1(\mu, x, s), s) \, \mathrm{d}\mu(x|s) \frac{1_{[0,t]}(s)}{\bar{\mu}([0,t])} \, \mathrm{d}\bar{\mu}(s)$$

$$= \int f(\Gamma_1(\mu_s, x, s), s) \, \mathrm{d}\mu(x|s) \frac{1_{[0,t]}(s)}{\bar{\mu}([0,t])} \, \mathrm{d}\bar{\mu}(s)$$

$$= \int f(\Gamma_1(\mu_t, x, s), s) \, \mathrm{d}\mu(x|s) \frac{1_{[0,t]}(s)}{\bar{\mu}([0,t])} \, \mathrm{d}\bar{\mu}(s)$$

$$= \int f(\Gamma_1(\mu_t, x, s), s) \, \mathrm{d}\mu_t(x, s)$$

$$= \int f(x, s) \, \mathrm{d} \left[ (\Gamma_1(\mu_t), \mathrm{Id}_{\mathbb{R}})_\sharp \mu_t \right](x, s),$$

where we have used that $\mu_s = \mu_t$ when $s \leq t$. This shows (36).

We see that by using the causality of $\Gamma_1$ and $\Gamma_2$, and (36)

$$\Gamma_2 \diamond \Gamma_1(\mu, x, t) = \Gamma_2 \left( (\Gamma_1(\mu), \mathrm{Id}_{\mathbb{R}})_\sharp \mu, \Gamma_1(\mu, x, t), t \right)$$

$$= \Gamma_2 \left( \left[ (\Gamma_1(\mu), \mathrm{Id}_{\mathbb{R}})_\sharp \mu \right]_t, \Gamma_1(\mu_t, x, t), t \right)$$

$$= \Gamma_2 \left( (\Gamma_1(\mu_t), \mathrm{Id}_{\mathbb{R}})_\sharp \mu_t, \Gamma_1(\mu_t, x, t), t \right)$$

$$= \Gamma_2 \diamond \Gamma_1(\mu_t, x, t).$$

These discussions apply for $t \in (0, 1]$, and the case when $t = 0$ follows by the same argument. Thus, we obtains the causality of $\Gamma_2 \diamond \Gamma_1$.

Assume that $\mu_t = \mu_{t'}$ where $\mu \in \mathrm{Lip}_C^\sigma(\tilde{\Omega})$ and $t, t' \in [0, 1]$. Without loss of generality, assume that $t < t'$. We have that by the identifiability of $\Gamma_1$ and $\Gamma_2$, and (36)

$$\Gamma_2 \diamond \Gamma_1(\mu_t, x, t) = \Gamma_2 \left( (\Gamma_1(\mu_t), \mathrm{Id}_{\mathbb{R}})_\sharp \mu_t, \Gamma_1(\mu_t, x, t), t \right)$$

$$= \Gamma_2 \left( (\Gamma_1(\mu_t), \mathrm{Id}_{\mathbb{R}})_\sharp \mu_t, \Gamma_1(\mu_{t'}, x, t'), t \right)$$

$$= \Gamma_2 \left( \left[ (\Gamma_1(\mu), \mathrm{Id}_{\mathbb{R}})_\sharp \mu \right]_t, \Gamma_1(\mu_{t'}, x, t'), t \right)$$

$$= \Gamma_2 \left( \left[ (\Gamma_1(\mu), \mathrm{Id}_{\mathbb{R}})_\sharp \mu \right]_{t'}, \Gamma_1(\mu_{t'}, x, t'), t' \right)$$

$$= \Gamma_2 \left( (\Gamma_1(\mu_{t'}), \mathrm{Id}_{\mathbb{R}})_\sharp \mu_{t'}, \Gamma_1(\mu_{t'}, x, t'), t' \right)$$

$$= \Gamma_2 \diamond \Gamma_1(\mu_{t'}, x, t'),$$

where we have used the following fact from (37)

$$\left[ (\Gamma_1(\mu), \mathrm{Id}_{\mathbb{R}})_\sharp \mu \right]_t = \left[ \Gamma_1(\mu)(\cdot, s)_\sharp \mu(\cdot|s) \right] \frac{1_{[0,t]}}{\bar{\mu}([0,t])} \bar{\mu}(s)$$

$$= \left[ \Gamma_1(\mu)(\cdot, s)_\sharp \mu(\cdot|s) \right] \frac{1_{[0,t']}}{\bar{\mu}([0,t'])} \bar{\mu}(s) = \left[ (\Gamma_1(\mu), \mathrm{Id}_{\mathbb{R}})_\sharp \mu \right]_{t'}.$$

These discussions apply for $t \in (0, 1]$, and the case when $t = 0$ follows by the same argument. Thus, we obtain the identifiability of $\Gamma_2 \diamond \Gamma_1$. $\qquad\square$

**Lemma 13.** *Identifiability is stable in the following sense: Let $\Lambda_n$ be continuous and causal, identifiable in-context mappings, and let $\Lambda^*$ be continuous and causal in-context mappings. Assume that, as $n \to \infty$,*

$$\sup_{(\mu, x, t) \in \mathrm{Lip}_C^\sigma(\tilde{\Omega}) \times \tilde{\Omega}} |\Lambda_n(\mu, x, t) - \Lambda^*(\mu, x, t)| \to 0. \tag{38}$$

*Then, the map $\Lambda^*$ is identifiable.*

*Proof.* Assume that $\mu_t = \mu_{t'}$ where $\mu \in \mathrm{Lip}_C^\sigma(\tilde{\Omega})$ and $t, t' \in [0, 1]$. As $\Lambda_n(\mu_t, x, t) = \Lambda_n(\mu_{t'}, x, t')$ and $\mu_t, \mu_{t'} \in \mathrm{Lip}_C^\sigma(\tilde{\Omega})$, we see that

$$|\Lambda^*(\mu_t, x, t) - \Lambda^*(\mu_{t'}, x, t')|$$

$$\leq |\Lambda^*(\mu_t, x, t) - \Lambda_n(\mu_t, x, t)| + |\Lambda_n(\mu_t, x, t) - \Lambda^*(\mu_{t'}, x, t')|$$

$$\leq \sup_{(\mu, x, t) \in \mathrm{Lip}_C^\sigma(\tilde{\Omega}) \times \tilde{\Omega}} |\Lambda^*(\mu, x, t) - \Lambda_n(\mu, x, t)| + |\Lambda_n(\mu_{t'}, x, t') - \Lambda^*(\mu_t', x, t')|$$

$$\leq 2 \sup_{(\mu, x, t) \in \mathrm{Lip}_C^\sigma(\tilde{\Omega}) \times \tilde{\Omega}} |\Lambda^*(\mu, x, t) - \Lambda_n(\mu, x, t)| \to 0,$$

which implies that

$$\Lambda^*(\mu_t, x, t) = \Lambda^*(\mu_{t'}, x, t').$$

$\square$

## C.2 PROOF OF LEMMA 4

For the representation (i) we find that, by using (20), (21) and $\mu_t = \mu_{e(\bar{\mu}_t)}$,

$$\Lambda(\mu, x, t) = \Lambda(\mu_t, x, t) = \Lambda(\mu_{e(\bar{\mu}_t)}, x, e(\bar{\mu}_t)) = \bar{\Lambda}(\mu_{e(\bar{\mu}_t)}, x) = \bar{\Lambda}(\mu_t, x),$$

where we have used, for the second and fourth equality, $\mu_t = \mu_{e(\bar{\mu}_t)}$.

The continuity (ii) follows from (20). Indeed, we observe that

$$\bar{\Lambda}(\mu_t, x) = \Lambda(\mu_t, x, e(\bar{\mu}_t)) = \Lambda(\mu_{e(\bar{\mu}_t)}, x, e(\bar{\mu}_t)). \tag{39}$$

Viewing

$$\mu_{e(\bar{\mu}_t)} = (\mu_{e(\bar{\mu}_t)})_{e(\bar{\mu}_t)} = (\mu_{e(\bar{\mu}_t)})_1 = \mu_t, \tag{40}$$

where $(\mu_{e(\bar{\mu}_t)})_{e(\bar{\mu}_t)}$ and $(\mu_{e(\bar{\mu}_t)})_1$ are regarded as the masked probability measures of $\mu_{e(\bar{\mu}_t)}$ at $t = e(\bar{\mu}_t)$ and $t = 1$, respectively, we obtain, using (20), (39) and (40), that

$$\bar{\Lambda}(\mu_t, x) = \Lambda((\mu_{e(\bar{\mu}_t)})_{e(\bar{\mu}_t)}, x, e(\bar{\mu}_t)) = \Lambda((\mu_{e(\bar{\mu}_t)})_1, x, 1) = \Lambda(\mu_t, x, 1).$$

Thus, by the continuity of $\Lambda$, we conclude that the map $(\mu_t, x) \mapsto \bar{\Lambda}(\mu_t, x) = \Lambda(\mu_t, x, 1)$ is continuous. $\square$

## C.3 PROOF OF LEMMA 5

Assume that $\mu_n \in \mathrm{Lip}_C^\sigma(\tilde{\Omega})$ and $(x_n, t_n) \in \tilde{\Omega}$. We see that

$$\{s \mapsto \mu_n(\cdot|s)\}_{n \in \mathbb{N}} \subset C([0, 1]; \mathcal{P}(\Omega)) \text{ is equicontinuous,}$$

as $s \mapsto \mu_n(\cdot|s)$ is $C$-Lipschitz. We also see that

$$\overline{\{\mu_n(\cdot|s)\}_{n \in \mathbb{N}}} \subset \mathcal{P}(\Omega) \text{ is compact for each } s \in [0, 1].$$

as $\mathcal{P}(\Omega)$ is compact in the $W_2$ topology (see e.g., Aliprantis & Border (2006, Theorem 15.11)). By the Arzelà–Ascoli theorem (Kelley, 2017, Chapter 7, Theorem 17), there exists $\mu(\cdot|s) \in \mathcal{P}(\Omega)$ such that the map $s \mapsto \mu(\cdot|s)$ is continuous map and (if needed, re-choose a subsequence)

$$\sup_{s \in [0,1]} W_2(\mu_n(\cdot|s), \mu(\cdot|s)) \to 0 \text{ as } n \to \infty. \tag{41}$$

As $(\bar{\mu}_n)_{n \in \mathbb{N}} \subset \mathcal{P}_\sigma([0, 1]) := \{\nu \in \mathcal{P}([0, 1]) : \nu(\{0\}) \geq \sigma\}$ and $\mathcal{P}_\sigma([0, 1])$ is compact, there exists $\bar{\mu} \in \mathcal{P}_\sigma([0, 1])$ such that (if needed, re-choose a subsequence) as $n \to \infty$

$$\bar{\mu}_n \rightharpoonup^* \bar{\mu}.$$

We set

$$\mu := \mu(\cdot|s)\bar{\mu}(s).$$

Then we have

$$\mu \in \mathrm{Lip}_C^\sigma(\tilde{\Omega}),$$

because

$$W_2(\mu(\cdot|s), \mu(\cdot|s')) \leq W_2(\mu(\cdot|s), \mu_n(\cdot|s)) + W_2(\mu_n(\cdot|s), \mu_n(\cdot|s')) + W_2(\mu_n(\cdot|s'), \mu(\cdot|s'))$$
$$\leq 2 \sup_{s \in [0,1]} W_2(\mu(\cdot|s), \mu_n(\cdot|s)) + C|s - s'|,$$

and taking limit as $n \to \infty$, we see that $s \mapsto \mu(\cdot|s) \in \mathcal{P}(\Omega)$ is $C$-Lipschitz.

Note that form (41)

$$\forall g \in \mathcal{C}(\Omega), \quad \sup_{s \in [0,1]} \left| \int g(x) \, \mathrm{d}\mu_n(x|s) - \int g(x) \, \mathrm{d}\mu(x|s) \right| \to 0. \tag{42}$$

Indeed, since the set $\mathrm{Lip}(\Omega)$ of all Lipschitz functions on $\Omega$ is dense in $\mathcal{C}(\Omega)$, for any $g \in \mathcal{C}(\Omega)$ and any $\epsilon \in (0,1)$, we choose $h \in \mathrm{Lip}(\Omega)$ such that $\sup_{x \in \Omega} |g(x) - h(x)| \leq \epsilon$. We see that as $W_1 \leq W_2$ and the dual formulae

$$\int g(x) \, \mathrm{d}\mu_n(x|s) - \int g(x) \, \mathrm{d}\mu(x|s)$$
$$\leq 2 \sup_{x \in \Omega} |g(x) - h(x)| + \mathrm{Lip}(h) \left( \int \frac{h(x)}{\mathrm{Lip}(h)} \, \mathrm{d}\mu_n(x|s) - \int \frac{h(x)}{\mathrm{Lip}(h)} \, \mathrm{d}\mu(x|s) \right)$$
$$\leq 2\epsilon + \mathrm{Lip}(h) W_1(\mu_n(\cdot|s), \mu(\cdot|s)) \leq 2\epsilon + \mathrm{Lip}(h) W_2(\mu_n(\cdot|s), \mu(\cdot|s)).$$

Taking $\sup_{s \in [0,1]}$ and the limit as $n \to \infty$, we obtain (42).

As $(x_n, t_n) \in \tilde{\Omega}$ and $\tilde{\Omega}$ is compact, there are $(x, t) \in \tilde{\Omega}$ (if needed re-choose the subsequence) such that

$$(x_n, t_n) \to (x, t) \text{ in } \tilde{\Omega}.$$

We finally need to show that, as $n \to \infty$,

$$(\mu_n)_{t_n} \rightharpoonup^* \mu_t,$$

which is equivalent to

$$\forall f \in \mathcal{C}(\tilde{\Omega}), \quad \int f \, \mathrm{d}(\mu_n)_{t_n} \to \int f \, \mathrm{d}\mu_t. \tag{43}$$

It is enough to check on any functions $f$ which are separable, i.e. of the form $f(x, s) = g(x)h(s)$ because linear combinations of separable functions of the form $\sum_i g_i(x)h_i(s)$ are dense in $\mathcal{C}(\tilde{\Omega})$.

To prove (43), we distinguish three cases (appropriately choosing a subsequence again):

(i) $t_n \in (0, 1]$ and $t \in (0, 1]$,    (ii) $t_n \in (0, 1]$ and $t = 0$,    and    (iii) $t_n = t = 0$

CASE (i): We see that as $n \to \infty$

$$\int f \, \mathrm{d}(\mu_n)_{t_n} - \int f \, \mathrm{d}\mu_t$$
$$= \int_{[0,1]} \frac{\mathbb{1}_{[0,t_n]}(s)h(s)}{\bar{\mu}_n([0,t_n])} \left( \int_\Omega g(x) \, \mathrm{d}\mu_n(x|s) \right) \mathrm{d}\bar{\mu}_n(s) - \int_{[0,1]} \frac{\mathbb{1}_{[0,t]}(s)h(s)}{\bar{\mu}([0,t])} \left( \int g(x) \, \mathrm{d}\mu(x|s) \right) \mathrm{d}\bar{\mu}(s)$$
$$\to 0,$$

because $\bar{\mu}([0,t]) > 0$, equation (42), and using that $\bar{\mu}_n \rightharpoonup^* \bar{\mu}$.

CASE (ii): We see that as $n \to \infty$

$$\int f \, \mathrm{d}(\mu_n)_{t_n} - \int f \, \mathrm{d}\mu_0$$

$$= \int_{[0,1]} \frac{\mathbf{1}_{[0,t_n]}(s)h(s)}{\bar{\mu}_n([0,t_n])} \left( \int_\Omega g(x) \, \mathrm{d}\mu_n(x|s) \right) \mathrm{d}\bar{\mu}_n(s) - h(0) \int g(x) \, \mathrm{d}\mu(x|0)$$

$$= \int_{[0,1]} \frac{\mathbf{1}_{[0,t_n]}(s)h(s)}{\bar{\mu}_n([0,t_n])} \left( \int_\Omega g(x) \, \mathrm{d}\mu_n(x|s) \right) \mathrm{d}\bar{\mu}_n(s) - \int_{[0,1]} \frac{\mathbf{1}_{[0,t_n]}(s)h(0)}{\bar{\mu}_n([0,t_n])} \left( \int g(x) \, \mathrm{d}\mu(x|0) \right) \mathrm{d}\bar{\mu}_n(s)$$

$$\leq \sup_{s \in [0,t_n]} \left| h(s) \int g(x) \, \mathrm{d}\mu_n(x|s) - h(0) \int g(x) \, \mathrm{d}\mu(x|0) \right| \left| \int_{[0,1]} \frac{\mathbf{1}_{[0,t_n]}(s)}{\bar{\mu}_n([0,t_n])} \, \mathrm{d}\bar{\mu}_n(s) \right|$$

$$\leq \sup_{s \in [0,t_n]} \left| h(s) \int g(x) \, \mathrm{d}\mu_n(x|s) - h(s) \int g(x) \, \mathrm{d}\mu(x|s) \right|$$

$$+ \sup_{s \in [0,t_n]} \left| h(s) \int g(x) \, \mathrm{d}\mu(x|s) - h(0) \int g(x) \, \mathrm{d}\mu(x|0) \right|$$

$$\leq \sup_{s \in [0,1]} |h(s)| \sup_{s \in [0,1]} \left| \int g(x) \, \mathrm{d}\mu_n(x|s) - \int g(x) \, \mathrm{d}\mu(x|s) \right|$$

$$+ \sup_{s \in [0,t_n]} \left| h(s) \int g(x) \, \mathrm{d}\mu(x|s) - h(0) \int g(x) \, \mathrm{d}\mu(x|0) \right| \to 0,$$

where we have used (42) and the continuity of the map $s \mapsto h(s) \int g(x) \, \mathrm{d}\mu(x|s)$.

CASE (iii): We see that, as $n \to \infty$,

$$\int f \, \mathrm{d}(\mu_n)_{t_n} - \int f \, \mathrm{d}\mu_t = \int f \, \mathrm{d}(\mu_n)_0 - \int f \, \mathrm{d}\mu_0$$

$$= h(0) \int g(x) \, \mathrm{d}\mu_n(x|0) - h(0) \int g(x) \, \mathrm{d}\mu(x|0)$$

$$\leq |h(0)| \left| \int g(x) \, \mathrm{d}\mu_n(x|0) - \int g(x) \, \mathrm{d}\mu(x|0) \right| \to 0,$$

by using (42). Therefore, we obtain (43).

# D   EXAMPLES OF OUR THEORY

## D.1   DISCRETE CASE

We consider a fixed $n$ and an in-context map $G(X, x)$, with $X \in \mathbb{R}^{d_{\text{tok}} \times n}$, which is continuous on $\ell^2$ and permutation equivariant with respect to the tokens. This defines a map on discrete probability measures

$$\Gamma\left( \frac{1}{n} \sum_i \delta_{x_i}, x \right) := G((x_i)_i, x),$$

which is continuous for the weak* topology on the set, $\mathcal{P}_n(\Omega) \subset \mathcal{P}(\Omega)$, of $n$-point measures (that is, uniform distributions supported on $n$ points). The map $\Gamma$ is continuous on $\mathcal{P}_n(\Omega)$ (because on point sets, the weak* topology coincides with the $\ell^2$-topology up to permutations), and $\mathcal{P}_n(\Omega)$ is compact (because it is a closed subset of a compact set $\mathcal{P}(\Omega)$). Hence, we can use our theorem on $\mathcal{P}_n(\Omega)$, and obtain that $\Gamma$ can be approximated by a transformer on this space. This implies the approximation of $G$ by a transformer.

## D.2   LINEAR REGRESSION

In the discrete case, tokens are assumed to be of the form $x_i = (u_i, v_i)$ where $u_i$ are features and $v_i$ are labels to be predicted. Then simplified (linear attention) transformers are shown to learn

in context a linear relation $v_i \approx W u_i$; the in-context (I-C) "prediction" then maps some $(u, v)$ to $(u, Wu)$ (that is, the value of $v$ is discarded). Adding a ridge penalty, $\lambda$, to make the problem well-posed, this corresponds to the I-C map

$$G(X, (u, v)) := (u, W(X)u) \quad \text{where} \quad W(X) := \text{argmin}_W \sum_{i=1}^{n} \|Wu_i - v_i\|^2 + \lambda\|W\|^2.$$

We note that Von Oswald et al. (2023) consider, in fact, a single attention layer and replace this minimization with a single step of gradient descent for simplicity; however, this is just a modification of the in-context map.

Thanks to our framework which operates over measures, the above regression can be written, for any $n$, upon considering a data distribution $\mu$ over the space $(u, v)$ of (feature, label), in terms of the more general in-context map,

$$\Gamma(\mu, (u, v)) := (u, W(\mu)u) \quad \text{where} \quad W(\mu) := \text{argmin}_W \int \|Wu - v\|^2 \mathrm{d}\mu(u, v).$$

This map has the closed form,

$$W(\mu) = \left[ \int uu^\top \mathrm{d}\mu(u, v) + \lambda \mathrm{Id} \right]^{-1} \left[ \int vu^\top \mathrm{d}\mu(u, v) \right],$$

and is weak* continuous as long as $\lambda > 0$. Hence, our theorem states that it can be learned in context.

