# OpenReview forum: "Transformers are Universal In-context Learners"
_ICLR.cc/2025/Conference — ICLR 2025 Poster_

### Official Review · Reviewer_d9aW · 2024-10-22

**Soundness:** 4
**Presentation:** 2
**Contribution:** 2
**Rating:** 6
**Confidence:** 3

**Summary:**

The paper considers the ability of transformers to approximate arbitrary in-context mappings, namely, their ability to transform an input set of token embeddings into an output set of embeddings arbitrarily close to a given target set, where each input token is transformed based on the context of the other input tokens. The input set is not limited to be a discrete finite set, but can be infinite or even continuous: in general it can be represented by a probability measure over the embedding space. A single transformer is able to obtain a given approximation level (aka precision), independently of the cardinality of the input set, which can even be infinite. The authors consider two settings: unmasked (aka non-causal, often used in vision applications) and masked (aka causal, often used in NLP), the masked setting requiring an extension of the approach used in the unmasked setting.

**Strengths:**

The paper explores in depth an interesting theoretical question about the in-context mapping abilities of the transformer architecture, namely its approximation abilities in presence of a context of arbitrary (even continuously infinite) cardinality.

The depth and rigour of the mathematical derivations appear (from the best judgment of a non-specialist, see also below) to be of very high quality.

**Weaknesses:**

The paper is extremely technical, and would require a considerable amount of time and effort, even by a mathematically inclined reader, to be fully understood.

The paper, as it stands, appears to be oriented towards a small number of readers, already familiar with the technical literature relating to measure-theoretic views of the approximation properties of transformers. Its appeal could be made broader by: (1) motivating the approach by potential (even non-immediate) applications and (2) by providing graphical illustrations of some of the underlying mathematical concepts, to help intuition (the current version of the paper does not provide a single illustration).

As the authors acknowledge, their results "are not quantitative", meaning (if I understand correctly), that a number of important parameters of the transformer architecture, such as the number of layers, are not bounded in terms of the approximation precision $\epsilon$. As they admit, this is a serious obstacle relative to applications. However, on the positive side, their current results provide some welcome clarification of the theoretical landscape.

The title "Transformers are Universal In-Context Learners" could be misleading to many readers, as it may seem to address the well-known question about the "ICL (In-Context Learning)" abilities of transformers, for example, the abilities of *fixed* LLMs, when provided with a "few-shot prompt of examples", to appear to "learn" to reproduce the "intention" expressed by the examples. In the current paper, the goal appears to be of a rather different nature, having to do with *finding* a transformer architecture and its parameters to approximate a given target contextual embedding. It would be useful if the authors could clarify whether they see a connection between their work and the usual notion of ICL.

**Questions:**

1 (Question). Could you provide one or two concrete examples that would illustrate the potential applicability of the approach? Such examples would improve the accessibility of the paper to a wider audience.

2 (Question). It is not fully clear to me why putting a probability measure over the input tokens is a natural move in the context of transformers. For instance, when considering a standard input of $n$ tokens, the usual representation is in terms of a set of cardinality $n$, and I am not quite sure why putting different weights over these tokens is an appropriate representation for modeling in-context mappings.

3 (Remark). On line 289, you say "Since our results are not quantitative, this is not a strong restriction." I had some difficulty making sense of this statement, starting with the unclarity of the term "not quantitative", which I finally understood to mean that you do not state bounds relative to $\epsilon$, e.g. concerning the number of layers $L$ (which could be mentioned along with the number of MLP parameters).

**Details Of Ethics Concerns:**

Thank you for your detailed responses. I have upgraded my rating to 6 and hope the paper will be accepted.

---

> ### Author Response · Authors · 2024-11-22
>
> We appreciate the detailed suggestions, criticisms and endorsement of the reviewer. We address these below:
>
> > (1) motivating the approach by potential (even non-immediate) applications
> > 1 (Question). Could you provide one or two concrete examples ... accessibility of the paper to a wider audience.
>
> We agree that we could have done a better job at giving some concrete example to illustrate the scope of our results.
> The first (and most important) example we have added to the manuscript is the one of a fixed number of tokens.
> This clearly show why discrete, fixed token length, is a special case of our theory.
> Namely, if we consider a fixed $n$ and an in-context map $G(X,x)$, with $X \in \mathbb{R}^{d_{\text{tok}} \times n}$ which is continuous for $\ell^2$ and permutation equivariant with respect to the token, then this defines a map on discrete probability measure
> $$
>     \Gamma(\frac{1}{n}\sum_i \delta_{x_i}, x) :=
>     G( (x_i)_i, x )
> $$
> which is continuous for the weak$^*$ topology  on the set $\mathcal{P}_n(\Omega) \subset \mathcal{P}(\Omega)$ of $n$-point measure (uniform distribution supported on $n$ points).
> The map $\Gamma$ is continuous on $\mathcal{P}_n(\Omega)$ (because on point sets, the weak$^*$ topology coincides with the $\ell^2$-topology up to permutations), and $\mathcal{P}_n(\Omega)$ is compact (because it is a closed subset of a compact set $\mathcal{P}(\Omega)$). Hence, we can use our theorem on $\mathcal{P}_n(\Omega)$, and obtain that $\Gamma$ can be approximated by a transformer on this space. This implies the approximation of $G$ by a transformer.
>
> The second example we now provide is a regression task associated with the “in-context learning” phenomenon, which we detail below to address your next question.
>
> > (2) by providing graphical illustrations ... does not provide a single illustration).
>
> We appreciate this suggestion. We will include a figure in the revised version.
>
>
> > The title "Transformers are Universal In-Context Learners" could be misleading to many readers ... clarify whether they see a connection between their work and the usual notion of ICL.
>
> We do somewhat disagree; we are using "in context" in the same way as in the recent theoretical literature on ICL. We, however, acknowledge that we do not study "learning'' mechanisms (we do not study the optimization of the $(Q,K,V)$) but rather state a "possibility'' (i.e. universality) result: it is possible to learn I-C mapping.
> To make this connection with the literature more concrete, we have updated the manuscript with the example of the  I-C linear regression task studied in [Johannes von Oswald et al., Transformers Learn In-Context by Gradient Descent, 2022].
> In the discrete case, tokens are assumed to be of the form $x_i=(u_i,v_i)$ where $u_i$ are feature and $v_i$ labels to be predicted. Then simplified (linear attention) transformers are shown to learn in context a linear relation $v_i \approx W u_i$ and the in-context "prediction'' then maps some $(u,v)$ to $(u, W u)$ (the value of $v$ is discarded).
> Adding a ridge penalty $\lambda$ to makes the problem well-posed, this corresponds to the I-C map
> $$
>     G(X,(u,v)) := (u, W(X) u)
>     \quad\text{where} \quad   W(X) := \mathrm{argmin}_{W} \sum_i  \|W u_i - v_i \|^2 + \lambda \|W\|^2
> $$
> (the authors of the paper consider in fact a single attention layer and replace this minimization with a single step of gradient descent for simplicity, but this is just a modification of the IC map).
> Thanks to our framework which operates over measure, this can be written for any $n$ by considering a data distribution $\mu$ over the space  $(u,v)$ of (feature, labels), and then defining the more general I-C map
> $$
>     \Gamma(\mu,(u,v)) := (u, W(\mu) u)
>     \quad\text{where}\quad
>     W(\mu) := \mathrm{argmin}_W \int  \|W u - v\|^2 \mathrm{d} \mu(u,v).
> $$
> This map has a closed-form
> $$
>     W(\mu) = \Big[ \int uu^\top \mathrm{d} \mu(u,v) + \lambda \mathrm{Id} \Big]^{-1} \Big[  \int vu^\top \mathrm{d} \mu(u,v)  \Big],
> $$
> and it is weak$^*$ continuous as long as $\lambda>0$, so our theorem states that it **can** be learned in context.

---

> > ### Author Response · Authors · 2024-11-22
> >
> > > 2 (Question). It is not fully clear to me why putting a probability measure over ... is an appropriate representation for modeling in-context mappings.
> >
> > We agree that the natural setup involves a discrete set of $n$ points. The primary motivation for extending this to probability distributions is to handle an arbitrary number $n$ of points, which requires introducing the weak$^*$ topology as $ n \to +\infty $.
> > As noted in response to the first question, this probability measure extension includes standard discrete transformers as a special case, specifically
> > $$
> > \mu = \frac{1}{n} \sum_{i=1}^{n}\delta_{x_i},
> > $$
> > where the number $ n $ of tokens is fixed.
> > This extension enables input of any token length, i.e., any $ n \in \mathbb{N} $, and in the universality result, the number of required transformer parameters remains independent of $ n $ (Theorem 1). This independence is one of the novel aspects of our work compared to existing approaches.
> > While this approach is not currently implemented in standard transformers, our extension also allows for consideration of weighted distributions of the form
> > $$
> > \mu = \sum_{i=1}^{n}a_i \delta_{x_i},
> > $$
> > where $ a_i > 0 $ and $\sum_i a_i = 1$. These weights could introduce a notion of uncertainty in the tokens. We have updated the manuscript to clarify these aspects.
> >
> > > 3 (Remark). On line 289, you say "Since our results are not quantitative, ... (which could be mentioned along with the number of MLP parameters).
> >
> > We agree that this sentence was unclear. By “non-quantitative,” we were referring to the lack of control over the number of layers. We acknowledge that saying “this is not an issue” was misleading. What we meant is that non-quantitative results naturally arise when only continuity is assumed (as is the case for two-layer MLPs). Obtaining quantitative results, however, would require imposing additional smoothness assumptions, and this appears to be a completely open problem for this type of mapping operating in finite dimensions (over the space of measures).

---

> > > ### Comment · Reviewer_d9aW · 2024-11-25
> > > **Thank you for your detailed responses.**
> > >
> > > Thank you for your detailed responses. I have upgraded my rating to 6 and hope the paper will be accepted.

---

### Official Review · Reviewer_9RCC · 2024-11-02

**Soundness:** 3
**Presentation:** 3
**Contribution:** 4
**Rating:** 6
**Confidence:** 2

**Summary:**

This paper examines Transformer's (unmasked and masked) ability to handle an unlimited number of context tokens. Contexts are modeled as probability distributions of tokens, with smoothness defined by continuity in Wasserstein distance. This paper shows that transformers can universally approximate continuous in-context mappings with fixed embedding dimensions and a constant number of heads.

**Strengths:**

I believe this paper provides a solid theoretical foundation for how a Transformer can model an infinite number of tokens, which supports the development of long-context language models. I find it especially surprising that a Transformer can handle an infinite number of tokens with a fixed embedding dimension.

**Weaknesses:**

I'm not sure if people actually use the unmasked variant discussed in this paper. For instance, bidirectional models like BERT and ViT must apply positional embeddings to their input tokens, meaning Eq. (2) isn’t typically used in practice. Therefore, I’d say the main contribution of this paper lies in its analysis of Eq. (3), which introduces additional regularity constraints on the token distribution.

**Questions:**

I don't have any specific question.

---

> ### Author Response · Authors · 2024-11-22
>
> We appreciate the detailed suggestions, criticisms and endorsement of the reviewer. We address all of these below:
>
> > I'm not sure if people actually use the unmasked ... regularity constraints on the token distribution.
>
> Classical transformers, such as BERT or ViT, indeed apply encoding to the tokens before passing them through the transformer layers. Therefore, they fit within our model by appropriately defining the $x_i$  to incorporate the embedding information. We have updated the manuscript to include this clarification. What remains open for future work is extending our approach to more recent positional encodings, such as Rotary Positional Embedding (RoPE) [Kazemnejad et al., The Impact of Positional Encoding on Length Generalization in Transformers, 2024]. RoPE modifies all attention layers to account for relative positional information, which would require slight adjustments to the proof to accommodate the different formulas.

---

> > ### Comment · Reviewer_9RCC · 2024-11-22
> > **Thank you for the response**
> >
> > Dear authors,
> >
> > Thank you for your response. I will retain my original score and lean toward accepting this paper.

---

### Official Review · Reviewer_5UVs · 2024-11-03

**Soundness:** 4
**Presentation:** 3
**Contribution:** 3
**Rating:** 8
**Confidence:** 2

**Summary:**

The paper proves that transformers can universally approximate context-dependent mappings, $G(\mathbf{x}; \mu)$. They do this by showing that one-dimensional attention functions are dense in the one-dimensional context-dependent mappings. Separate proofs are provided for masked and unmasked attention.

**Strengths:**

*Nice prior-work section, succinctly summarizes a very large literature.

*The "in-context mapping" formulation is nice, not sure if it was used in prior theory but seems to apply to many IC learners like RNNs and SSMs and so on.

*Feel like many of the techniques used could become useful theoretical constructions themselves (like the Laplace-like transform of Lemma 1)

**Weaknesses:**

Hard to think of any beyond those already acknowledged by the authors. If I were to reach for one, I feel like the result itself is maybe less interesting than the methods (which are very interesting), since being able to approximate any function often doesn't translate to being able to learn it (e.g. shallow networks, polynomial regression), and so it might be nice to spend some space in the intro or discussion highlighting what if any bearing this has on learning.

**Questions:**

Only what I mentioned above.

---

> ### Author Response · Authors · 2024-11-22
>
> We appreciate the detailed suggestions, criticisms and endorsement of the reviewer. We address all of these below:
>
> > Hard to think of any beyond those ... if any bearing this has on learning.
>
> We acknowledge that our results do not directly translate into conclusions about the learning capabilities of transformers. Our techniques, particularly the use of the Weierstrass theorem combined with Optimal Transport, bear similarities to methods employed in analyzing the training dynamics of two-layer MLPs, such as in [Chizat, Bach, On the Global Convergence of Gradient Descent for Over-parameterized Models using Optimal Transport, 2018]. Consequently, we believe that future research could extend our approach to explore convergence results for transformers. We have added further comments to elaborate on this point. That said, unlike classical MLPs with cosine activations, shallow architectures lack an algebraic structure (e.g., they cannot be multiplied), which makes the proof technique itself significant. To address this challenge, we relied on depth to overcome the limitation. We have also included additional remarks on the broader implications of the proof technique.

---

### Meta-Review · Area_Chair_2Gi8 · 2024-12-23

**Metareview:**

This paper presents a compelling advancement in the theoretical understanding of Instance-wise Contrastive Learning (ICL) within the context of Transformer models.  The reviewers unanimously agree that the paper makes a substantial contribution to the problem.

During the discussion phase, the authors diligently addressed the reviewers' comments and suggestions, further strengthening the paper. The revised manuscript effectively incorporates these improvements, enhancing clarity and addressing any initial concerns. In light of the paper's strong theoretical foundation, its positive reception by the reviewers, and the authors' responsiveness to feedback, I suggest acceptance.

**Additional Comments On Reviewer Discussion:**

Reviewers mainly raised clarification questions about the draft. Authors updated the draft to make the writing more crisp.

---

### Decision · Program_Chairs · 2025-01-22

Accept (Poster)